# The Role of Feature Interactions in Graph-based Tabular Deep Learning

**Elias Dubbeldam**                                    *e.f.dubbeldam@uva.nl*
*University of Amsterdam*

**Reza Mohammadi**                                    *a.mohammadi@uva.nl*
*University of Amsterdam*

**Marit Schoonhoven**                                    *m.schoonhoven@uva.nl*
*University of Amsterdam*

**Ilker Birbil**                                    *s.i.birbil@uva.nl*
*University of Amsterdam*

**Reviewed on OpenReview:** *https://openreview.net/forum?id=olGaiwoZHZ*

## Abstract

Accurate predictions on tabular data rely on capturing complex, dataset-specific feature interactions. Attention-based methods and graph neural networks, referred to as graph-based tabular deep learning (GTDL), aim to improve predictions by modeling these interactions as a graph. In this work, we analyze how these methods model the feature interactions. Current GTDL approaches primarily focus on optimizing predictive accuracy, often neglecting the accurate modeling of the underlying graph structure. Using synthetic datasets with known ground-truth graph structures, we find that current GTDL methods fail to recover meaningful feature interactions, as their edge recovery is close to random. This suggests that the attention mechanism and message-passing schemes used in GTDL do not effectively capture feature interactions. Furthermore, when we impose the true interaction structure, we find that the predictive accuracy improves. This highlights the need for GTDL methods to prioritize accurate modeling of the graph structure, as it leads to better predictions.

## 1 Introduction

Deep learning has achieved remarkable success in domains such as natural language processing and computer vision. On tabular data, however, deep learning methods still struggle to compete against traditional, tree-based machine learning methods (Grinsztajn et al., 2022; McElfresh et al., 2024). Although recent advances in tabular deep learning occasionally surpass these baselines on select benchmarks (e.g., Gorishniy et al. (2021); Hollmann et al. (2025)), no deep learning method has yet demonstrated consistent superiority across datasets and evaluation settings (Grinsztajn et al., 2022; McElfresh et al., 2024).

Tabular data is characterized by the heterogeneous nature of its features: each feature often encodes distinct semantics, and relationships among features (or feature interactions) can be complex, indirect, and dataset-specific. By modeling the feature interactions, one incorporates the *inductive bias* (domain-specific principles embedded into the model's architecture (Goyal & Bengio, 2022; Battaglia et al., 2018; Romero Guzman, 2024)) that features interact with each other differently. With 'modeling' we mean that the network has, by design, separate parameters for each feature interaction. Using inductive biases has proven to be important for success in other fields of deep learning. For example, convolutional neural networks (CNNs) achieve sample efficiency and robustness in computer vision by encoding translational invariance (LeCun et al., 1989; Fukushima, 1980), and transformers excel in natural language processing using attention-based mechanisms to capture sequential and contextual relationships (Vaswani et al., 2017).

Modeling this inductive bias of feature interactions comes naturally in the form of a graph, where the nodes represent features and the edges their interactions. Probabilistic graphical models (PGMs) have a rich history in statistics, providing a framework to model multivariate dependencies (Lauritzen, 1996). These methods excel at robustly describing the graph structure while enabling predictions, yet they lack the ability to model complex nonlinear relationships that deep learning can provide. Graph-based tabular deep learning (GTDL) methods aim to merge the expressive power of deep learning with graph-structured feature representations. *Feature* graph neural networks (GNNs), reviewed by Li et al. (2024), are GNNs focused on tabular data, having the features as nodes and the feature interactions as edges.[1] However, how these feature interactions are modeled within these networks has not been extensively studied or evaluated. These methods do not evaluate explicitly whether their learned feature interactions accurately correspond to meaningful relationships in the data.

Existing GTDL methods (e.g., Li et al. (2019a); Yan et al. (2023); Zheng et al. (2023); Villaizán-Vallelado et al. (2024); Ye et al. (2024); Zhou et al. (2022)) typically evaluate the learned graph structure only qualitatively, as real-world datasets rarely include ground-truth feature interaction graphs. Their training loss is tied to predictive performance, providing no incentive to ensure accuracy or meaningfulness in the learned graph structure. As a result, the adjacency matrix may reflect optimization artifacts rather than genuine feature interactions, as sketched in Figure 1. This emphasis on predictive metrics over structural fidelity constrains interpretability. Overall, there remains a lack of systematic techniques for both validating learned feature interactions and guiding learning with prior knowledge.

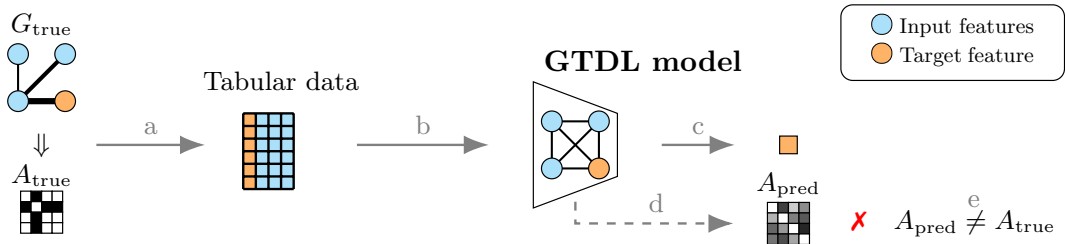

Figure 1: (a) A true underlying graph structure generates tabular data. (b, c) Only the data is used to train a GTDL model (using a fully connected graph) to predict the target feature. (d) After training, the learned graph structure is extracted from the model. (e) When this predicted graph structure is compared to the true graph structure, we find that they are not similar for existing GTDL methods.

In this work, we analyze whether existing GTDL methods learn meaningful feature interactions. Do GTDL methods learn an accurate graph structure when being trained to predict a target feature? Conversely, does an exclusive focus on predictive accuracy lead to spurious interactions rather than genuine feature dependencies, thereby undermining robustness, generalization, and explainability? When the graph structure is modeled correctly, does the predictive performance of GTDL methods improve?

To support our analysis, we take the following steps. In Section 2, we review the existing literature on GTDL methods and identify their limitations in the evaluation and validation of learned feature interactions. In Section 3, we introduce a framework to systematically evaluate how well predictive GTDL methods learn the graph structure. The two key parts of this framework are (i) synthetic datasets with known ground-truth graphs, and (ii) a metric to quantitatively assess the accuracy of learned graphs. In Section 4, we discuss the results of controlled experiments to showcase the framework. Specifically, we show that existing GTDL methods fail to recover meaningful feature interactions, and that enforcing the true interaction structure improves predictive performance. In Section 5, we give a concluding discussion and propose future research directions.

---

[1]This categorization of feature graphs contrasts with instance graphs, where nodes represent instances (rows) and edges capture relationships between those instances.

## 2 GTDL methods and their feature interactions

After formalizing the problem context, we organize this section around methods that model feature interactions in tabular data through feature graphs. First, we review attention-based methods and GNNs for tabular data, and discuss how both are interpreted as GTDL methods. Next, we relate GTDL to PGMs, which serve as a principled baseline for learning conditional independence structure. Lastly, we investigate methods that may appear related at first glance but, upon closer examination, are not directly pertinent to this study.

**Problem setting and notation.** A full tabular dataset $D = [x \parallel y] \in \mathbb{R}^{n \times p}$ consists in a traditional supervised setting of input features $x \in \mathbb{R}^{n \times (p-1)}$ and a target feature $y \in \mathbb{R}^{n \times 1}$, with $n$ the number of samples, $p$ the number of features and $\parallel$ indicating concatenation. When we refer to 'features', we mean both input and target features. Features could be either numerical or categorical.

The features and their interactions can be represented as an undirected graph $G = (V, E)$, where $V$ is the set of nodes and $E$ the set of edges, such that the number of nodes $|V| = p$. A binary symmetric adjacency matrix $A_{\text{true}}$ describes the true graph structure. The absence of an edge between two nodes indicates the conditional independence of these two nodes conditioned on all other nodes (this is different from directed causal graphs, where the presence of an edge indicates a direct causal effect between two nodes.)

From trained GTDL methods, we can extract a weighted adjacency matrix $A \in \mathbb{R}^{p \times p}$, where $0 \leq A_{ij} \leq 1$ indicates the strength of the interaction between features $i$ and $j$, with $i, j \in \{1, \ldots, p\}$. As the interaction from feature $i$ to $j$ could be different from the interaction from feature $j$ to $i$, the weighted adjacency matrix is not necessarily symmetric: While $A_{ij}$ can differ from $A_{ji}$, they should be symmetric in support (i.e., they share the same sparsity pattern). Specifically, if $A_{\text{true},ij} = 0$, then it should be that $A_{ij}, A_{ji} = 0$. Conversely, if $A_{\text{true},ij} = 1$, then it should be that $A_{ij}, A_{ji} > 0$.

The main task is to predict the target feature $y$ given the input features $x$. Correct modeling of the feature interactions could improve this prediction, including both target-input interactions (relationships between $y$ and each $x_i$) and input-input interactions (relationships between pairs of input features $x_i$ and $x_j$). The strength of feature interactions between two nodes should be (close to) zero when these nodes are conditionally independent.

### 2.1 Attention-based methods

Due to the success of the transformer architecture (Vaswani et al., 2017), most recent tabular deep learning methods are attention-based (Arik & Pfister, 2020; Huang et al., 2020; Somepalli et al., 2021; Kossen et al., 2021; Gorishniy et al., 2021), from which FT-Transformer (Gorishniy et al., 2021) has been established as a popular baseline. All of these methods are based on multi-head self-attention (Vaswani et al., 2017).

In most works, the attention map is of size $p \times p$ if a target token (or CLS, 'classification' or 'output' token (Devlin et al., 2019)) is appended. If the attention map is equal to the size of the total number of features (or the number of input features), it can be used for interpretation and explaining the feature interactions. This approach is popular in natural language, with BertViz (Vig, 2019) being used to interpret how the model assigns weights to different tokens. While in natural language the size of attention map differs per input due to varying sequence lengths, in tabular data the number of features is fixed, making the attention map the same size as the adjacency matrix. This allows for an interpretation of the attention map as a proxy for the weighted adjacency matrix, which is further discussed in Subsection 3.2. Therefore, we refer to such attention-based methods as *implicit* GTDL methods. They do not explicitly model the feature interactions, but due to the nature of the attention map, they do model the graph structure implicitly.

This notion, that the attention map can be interpreted as the learned graph structure of tabular data, has not been thoroughly discussed in the literature. However, most methods, as listed in Table 1, use the attention map for interpretability. TabNet (Arik & Pfister, 2020) reports the attention map of synthetic datasets as visualizations and notes how irrelevant features are ignored in the attention map. SAINT (Somepalli et al., 2021), although designed for tabular data, reports the attention map of MNIST and discusses that the visualization of the attention map is similar to the ground-truth image. FT-Transformer (Gorishniy et al., 2021) interprets the attention map as feature importances, and shows for some real-world datasets that the

attention map has a high rank correlation with integrated gradients (Sundararajan et al., 2017), a method to measure feature importance.

Table 1: GTDL methods evaluate the feature interaction only qualitatively, typically with a visualization of the attention map or adjacency matrix. Size denotes the length of square attention map or adjacency matrix $A$ (e.g., size $p$ means $A \in \mathbb{R}^{p \times p}$).

| Model | Reference | Size | Feature interaction evaluation |
|-------|-----------|------|-------------------------------|
| **Attention-based** | | | |
| FT-Transformer | Gorishniy et al. (2021) | $p$ | Correlation with feature importance |
| TabNet | Arik & Pfister (2020) | $p$ | Visual of synthetic dataset |
| SAINT | Somepalli et al. (2021) | $p$ | Visual of MNIST |
| **Graph neural network** | | | |
| FiGNN | Li et al. (2019a) | $p-1$ | Visual of real-world dataset |
| T2G-Former | Yan et al. (2023) | $p$ | Visual of real-world datasets |
| DRSA-Net | Zheng et al. (2023) | $p-1$ | Visual of real-world dataset |
| INCE | Villaizán-Vallelado et al. (2024) | $p$ | Visual of real-world dataset |
| MPCFIN | Ye et al. (2024) | $p$ | Visual of real-world datasets |
| Table2Graph | Zhou et al. (2022) | $p-1$ | Visual synthetic dataset |

## 2.2 Graph neural networks

GNNs operate directly on graph-structured data by propagating information between connected nodes (Zhou et al., 2021). Feature GNNs apply this paradigm to tabular data, modeling each feature as a node and explicitly learning feature interactions through message passing (Li et al., 2024).

Generally, the message-passing mechanism can be summarized as

$$h_i^{(l)} = \phi^{(l)} \left( h_i^{(l-1)}, \bigoplus_{j \in \mathcal{N}(i)} \psi^{(l)} \left( h_i^{(l-1)}, h_j^{(l-1)}, A_{ij} \right) \right), \tag{1}$$

with $h_i^{(l)}$ the representation of node $i$ at layer $l$, $\mathcal{N}(i)$ the neighbors of node $i$, [2] $\psi^{(l)}$ a message function, $\bigoplus$ an aggregation operator, and $\phi^{(l)}$ an update function. The details of these differ between GNN architectures. Attention-based methods can be seen as a special case of GNNs (Joshi, 2025). In the context of tabular data, a key advantage of GNNs is that they generalize attention-based methods by using a trainable weighted adjacency matrix, $A$, to explicitly propagate information between nodes—rather than relying on implicitly learned attention maps. Furthermore, GNNs for tabular data typically have trainable parameters that represent individual features or interactions, allowing for more flexible modeling of feature interactions compared to attention-based methods that rely on shared parameters across features. Therefore, we refer to them as *explicit* GTDL methods, contrary to attention-based methods that model the graph structure implicitly.

Different GTDL methods implement Equation (1) in different ways. FiGNN (Li et al., 2019a) uses a feature graph to explicitly model the separate feature interactions. T2G-Former (Yan et al., 2023) adapts the transformer architecture (Vaswani et al., 2017) for tabular data and learns a feature graph that focuses on learning meaningful interaction between different features. DRSA-Net (Zheng et al., 2023) uses dual-route structure GNNs to learn adaptively the sparse graph structure. INCE (Villaizán-Vallelado et al., 2024) has a similar approach as T2G-Former, but uses an Interaction Network (Battaglia et al., 2016) instead of a Transformer. MPCFIN (Ye et al., 2024) uses cross-feature embeddings and multiplex GNNs to model the interaction and dependencies between features.

---

[2]In GTDL, the neighbors are typically all other features, i.e., $\mathcal{N}(i) = \{1, \dots, p\} \setminus \{i\}$. A fully connected graph is used due to the absence of a known graph structure.

The feature GNN literature (Li et al., 2019a; Yan et al., 2023; Zheng et al., 2023; Villaizán-Vallelado et al., 2024; Ye et al., 2024; Zhou et al., 2022) suggests that the learned adjacency matrix can be used to interpret and explain the feature interactions. However, there are two reasons to be careful with this interpretation.

1. The evaluation of the graph structure is only heuristic to the best of our knowledge. The afore-mentioned explicit GTDL methods (FiGNN, INCE, T2G-Former, MPCFIN, and DRSA-Net) report the learned adjacency matrix for one or a few real-world datasets. They argue that the learned feature interactions are meaningful by post-hoc explaining the feature interactions. They justify the connections by referencing the semantic meaning of the feature names, suggesting that connected features are intuitively related. The issue with this approach is that, in the absence of a ground-truth graph structure, it becomes impossible to quantitatively evaluate the learned adjacency matrix.
2. The GTDL methods do not explicitly instruct the model to learn the true underlying graph structure. The loss is computed exclusively on the error between predicted and true target values. This means the model is only incentivized to improve predictive accuracy, not to accurately model the underlying feature interactions. As a result, the learned graph structure may not reliably reflect the true relationships between features. This limits its usefulness for interpretability and potentially constrains predictive performance.

Table2Graph (Zhou et al., 2022) addresses the first problem, that real-world datasets do not have a ground truth graph structure, by using a synthetic dataset. However, the learned graph structure is still evaluated heuristically against the ground-truth interactions, by visually comparing the learned weighted adjacency matrix with the ground-truth interactions. The second problem, that the model is only prediction-centric, is addressed by introducing a reinforcement learning term to the loss function to explore the adjacency matrix. This encourages the model to also focus on learning the graph structure, rather than just the predictive performance.

Not all GNN methods treat predicting the target feature in the same way. Some of these models treat predicting the target feature as a node-level task, while others treat it as a graph-level task (Prince, 2023). Node-level approaches (T2G-Former, INCE, MPCFIN) include a target node in the graph structure, and pass the embedding of the target node to an output layer. With this approach, the model learns a weighted adjacency matrix of size $p \times p$. Graph-level approaches (FiGNN, DSRA-Net, Table2Graph) do not include a target node, resulting in an adjacency matrix of size $(p-1) \times (p-1)$. The embeddings of all nodes are aggregated and passed to an output layer.

## 2.3 Probabilistic graphical model as a baseline for GTDL

Feature graphs, as studied by GTDL methods, share similarities with PGMs (Lauritzen, 1996; Koller & Friedman, 2009). PGMs provide a principled framework for modeling multivariate dependencies by encoding conditional independence relationships among random variables using graphs. Widely applied in Bayesian statistics, PGMs represent the structure of a probability distribution, often Gaussian, through a compact graph encoding the conditional independencies among variables. Bayesian techniques in PGMs, like BDgraph (Mohammadi & Wit, 2019), have demonstrated strong empirical performance in recovering interaction structures (Vogels et al., 2024). The ability to quantify uncertainty in the learned graph structure makes them a useful sanity check for GTDL methods.

## 2.4 Related approaches

There are other related approaches that are after closer inspection not relevant to our discussion on GTDL. In the literature on recommender systems and click-through rate, there has been a longer interest in a different notion of feature interactions, that of *cross features*; e.g., (Cheng et al., 2016; Guo et al., 2017; Lian et al., 2018; Wang et al., 2017; Cai et al., 2021; Wang et al., 2020; Song et al., 2019). These models focus on learning multiple weighted products of features to improve the prediction of the target feature. As noted by Li et al. (2019a), this limits the capability to model interactions across different features flexibly and explicitly. We are interested in how to model the feature interactions explicitly on a graph. Therefore, we do not discuss the feature interactions of these methods in further detail.

Tabular foundation models, TabPFN (Hollmann et al., 2025), TabICL (Qu et al., 2025) and LimiX (Zhang et al., 2025) have recently gained attention due to their high predictive performance on tabular data. Two key aspects of these models are their alternating instance-wise and feature-wise attention layers, and their ensembling predictions over multiple feature permutations. The feature-wise attention layers work similar as the attention-based methods discussed in Subsection 2.1, and therefore could have been interpreted as implicit GTDL methods. However, these tabular foundation models contain techniques that prevent a straightforward interpretation of the feature-wise attention map as a weighted adjacency matrix. TabPFN and LimiX encode groups of features collectively rather than individually. TabICL incorporates rotary positional embedding (RoPE) (Su et al., 2024) independent of the feature permutation, which alters the attention map. Therefore, we do not consider the feature interactions of these models in this work.

Tree-based models (e.g., XGBoost (Chen & Guestrin, 2016), LightGBM (Ke et al., 2017) and CatBoost (Prokhorenkova et al., 2019)) remain a popular choice for tabular data. Nevertheless, these approaches do not explicitly represent feature interactions in a graphical format. Due to the nature of tree architectures, the learned feature interactions can not easily be extracted from the model.

## 3 Evaluating feature interactions in GTDL

The development of GTDL is hindered by the fact that the learned graph structure is only evaluated heuristically. To solve this, we introduce a framework to evaluate the learned graph structure of GTDL methods with synthetic datasets and quantitative metrics.

### 3.1 Synthetic data with a graph structure

Most existing GTDL methods lack rigorous evaluation of the learned graph structure. Typically, the learned graph structure is evaluated heuristically, by reporting a visualization of the learned weighted adjacency matrix of real-world datasets. Feature interactions are post-hoc explained based on the semantic meaning of the feature names. Evaluating only on real-world datasets is problematic, as the *true* graph structure is not known. Therefore, we propose using *synthetic* datasets. Using synthetic data enables GTDL methods to compare the learned graph structure with the ground-truth underlying graph structure in a controlled environment.

We adapt two existing data generation methods from the literature. The three-step process is sketched in Figure 2, and the details are given in Appendix B.

- **Multivariate normals (MVNs)** are typically studied by PGM methods. We follow the default procedure of generating conditional multivariate data (as described in (Mohammadi & Wit, 2015), for instance). In short, we employ the following: (i) Sample a graph structure $G_{\text{true}} \in \mathbb{R}^{p \times p}$ from the Bernoulli distribution. (ii) Sample a covariance matrix $\Sigma_G \in \mathbb{R}^{p \times p}$ from the G-Wishart distribution (Roverato, 2002; Letac & Massam, 2007) to describe the feature interactions. (iii) Obtain $n$ samples $D \in \mathbb{R}^{n \times p}$ from $\mathcal{N}(0, \Sigma_G)$.

- **Structural causal model (SCM)** (Pearl, 2021) are used to generate tabular data in tabular foundation models (Hollmann et al., 2025; Qu et al., 2025; Zhang et al., 2025). We follow a simplified version of the data generation process in these tabular foundation models, that is: (i) Generate a directed acyclic graph (DAG) to define the graph structure of the SCM. To obtain the undirected graph $G_{\text{true}}$ from the DAG, we moralize (connect all parents of a child node, and drop the direction of the edges because the child of a parent can also be used to predict the parent) and marginalize (drop the root nodes and connect its children) the DAG (Cowell et al., 1999). More details on moralization and marginalization are discussed in Appendix B. (ii) Sample computational maps $f_i$ for each child node $i$ in the DAG. The computational maps $\{f_i\}$ are smooth nonlinear functions that take all the values of the incoming edges as input, and the output is the value of the child node $i$. (iii) Traverse random data $x_{\text{roots}}$ in topological order through the DAG to obtain $n$ samples $D \in \mathbb{R}^{n \times p}$.

For both approaches, we randomly select a target feature $y \in \mathbb{R}^{n \times 1}$ from $D$, with the remaining columns serving as input features $x \in \mathbb{R}^{n \times (p-1)}$. The target feature is directly influenced by its neighbors and only indirectly by non-neighbors. This setup lends itself well for evaluating the graph structure learned by GTDL

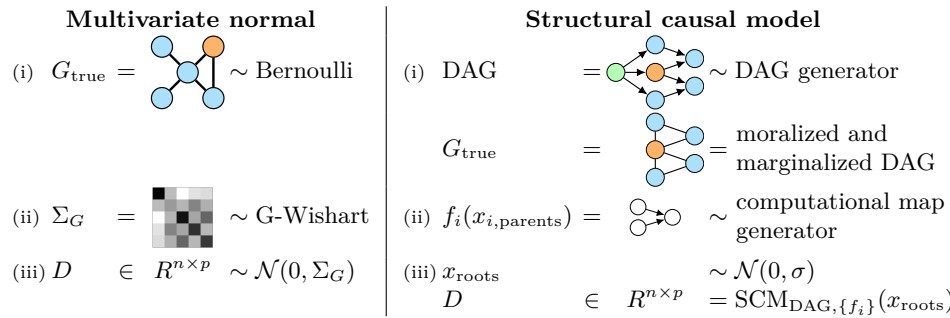

Figure 2: Two synthetic data generation pipelines. Both pipelines can roughly be divided into three steps. (i) Sample a graph structure. (ii) Sample feature interaction. (iii) Sample data given the graph and feature interactions. Nodes are colored as • cyan input features $x$, • orange target feature $y$, and • green root nodes $x_{\text{roots}}$.

methods. Consider the simple example: $x_0 - x_1 - x_2$. The model could learn to use $x_0$ to predict $x_2$ directly by learning an edge between them. However, this is suboptimal because $x_2$ is conditionally independent of $x_0$ given $x_1$. That is, once $x_1$ is known, $x_0$ provides no additional information for predicting $x_2$ (Lauritzen, 1996). When observations are noisy, using $x_0$ to predict $x_2$ accumulates uncertainty across multiple dependencies, leading to worse predictions than those obtained by using the immediate neighbor $x_1$. The model should instead recognize $x_1$ as a more reliable predictor for $x_2$, which results in learning the correct graph structure.

The proposed synthetic data generation methods are different from the synthetic datasets used by TabNet (Arik & Pfister, 2020) or Table2Graph (Zhou et al., 2022), previously discussed in Subsections 2.1 and 2.2. All input features in these datasets are conditionally independent of each other, and have a direct interaction with the target feature. Therefore, there is no underlying graph structure to be learned, as all features are connected only to the target feature. In contrast, the MVN and SCM data generation methods create datasets with more complex underlying graph structures, making them more suitable to evaluate GTDL methods.

We acknowledge that our data generation processes may not be fully representative of real-world tabular data. For instance, the graphs evaluated in this work (Subsection 3.4 and Section 4) are relatively small ($p = 10$), the MVN does only have linear feature interactions, and the SCM does not have missing nodes within the DAG. However, key is that the datasets have a clear underlying ground-truth graph structure that GTDL methods should be able to learn. If models can not model the feature interactions of these synthetic datasets well, it is unlikely to expect that, on larger ($p \gtrsim 100$) and real-world datasets, these models will learn meaningful feature interactions.

## 3.2 Interpreting and evaluating the graph structure

The GTDL methods introduced in Subsections 2.1 and 2.2 model the graph structure inside their architecture. In this section, we discuss how we extract the learned weighted adjacency matrix $A \in \mathbb{R}^{p \times p}$ from these methods, and how we propose to evaluate the quality of the learned graph structure. For the GNN-based GTDL methods, extracting the adjacency matrix is straightforward, as these methods explicitly use a weighted adjacency matrix within their architecture. For the attention-based GTDL methods, we interpret the attention maps as a proxy for the learned weighted adjacency matrix.

To justify this interpretation, consider the following example: Let $a$ be a $p \times p$ matrix representing the attention map, where each element $a_{ij}$, denotes the attention weight from feature $i$ to feature $j$, as used by Vaswani et al. (2017). If feature $i$ is conditionally independent of feature $j$ given all other features, we have $A_{\text{true},ij} = 0$. The attention mechanism should learn during training to assign low attention weights $a_{ij}, a_{ji} \approx 0$, as feature $i$ does not rely on information from feature $j$ (and vice versa) for its representation. Contrary, if feature $i$ and feature $j$ are dependent, we have $A_{\text{true},ij} = 1$. The attention mechanism should learn to assign non-zero attention weights $a_{ij}, a_{ji} > 0$, as feature $i$ relies on information from feature $j$ (and

vice versa) for its representation. In summary, with $A_{\text{true},ij} = 0$ we expect $a_{ij}, a_{ji} \approx 0$, and with $A_{\text{true},ij} = 1$, we expect $a_{ij}, a_{ji} > 0$.

After training, the attention maps of the test samples are extracted for all heads and layers. To obtain the adjacency matrix we perform two steps. First, we average the attention maps over the test samples, heads and layers to obtain and to obtain a single attention map of size $p \times p$. Second, we account for the diagonal of the attention map and for the softmax normalization from the original attention equation, which is further explained in Appendix A, to obtain the weighted adjacency matrix $A$.

Current GTDL methods only report the predictive performance of the target feature, and do not evaluate the learned feature interactions quantitatively. We propose to evaluate the quality of the graph structure by comparing edge-wise (ignoring the diagonal) the true binary adjacency matrix $A_{\text{true}}$ with the learned weighted adjacency matrix $A_{\text{pred}} = A$ with the receiver operating characteristic area under curve (ROC AUC) (Bradley, 1997). This metric reflects to what degree the feature interaction strengths of the true edges are higher than those of the true non-edges. Ranging from zero to one, a value of 0.5 equals a random guess. A high ROC AUC indicates that all true edges have higher feature interaction strengths than all true non-edges. The ROC AUC is a 'relative measure', meaning that it is not sensitive to the scale of the feature interaction strengths. This way, we are forgiving in the evaluation of the feature interactions, as we only measure if the model can distinguish between true edges and true non-edges, and not the absolute values of the feature interaction strengths.

### 3.3 Pruning the feature interactions

To understand the effect of learning the correct graph structure, we model the GTDL methods in two different settings, which are sketched in Figure 3. First, we train the GTDL with a fully connected graph. This is the default setting in GTDL methods, as the true graph in real-world datasets is not known. Second, we limit feature interactions to only those present in the synthetic data, effectively *pruning the graph to the true edges*. This means that the model is only allowed to learn feature interactions that are present in the true graph. Practically, this is done by masking the attention map or the graph structure within the network architecture. This is only possible if the true graph is known, which is the case for synthetic data. By comparing the results from the fully connected and the pruned graphs, we can see how much the GTDL methods benefit from using only the true edges.

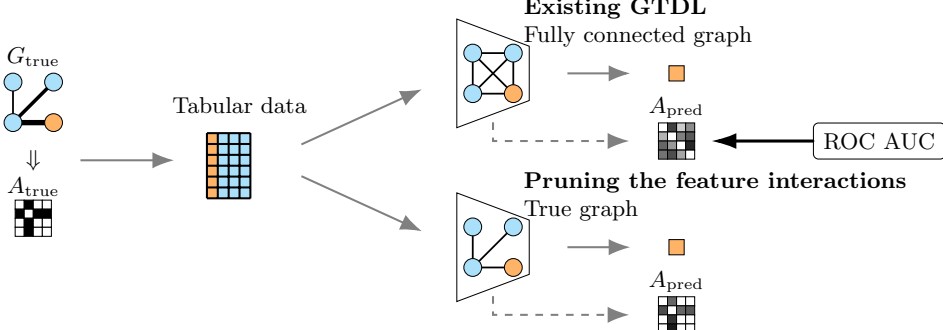

Figure 3: Upper branch (similar to Figure 1): Existing GTDL methods use a fully connected graph. The learned adjacency matrix $A_{\text{pred}}$, that we extract out of GTDL methods, is compared to the true adjacency matrix $A_{\text{true}}$ with the ROC AUC to evaluate the learned feature interactions. Lower branch: When the graph is pruned to the true edges, GTDL methods can only model feature interactions that are present in the true graph. By doing this, we can analyze the effect when the correct graph structure is used in GTDL methods.

### 3.4 Setup of the experiments

We conduct a standard deep learning experiment to evaluate existing GTDL methods. That is, we optimize the prediction of the target feature $y$ given the input features $x$. This is done with the default GTDL setting of a fully connected graph, and with the pruned setting where the feature interactions are limited to the true edges only. We tune the hyperparameters of the model and the learning rate, we use cross validation, and optimize the mean squared error (MSE) with Adam (Kingma & Ba, 2017). Further details on the splitting, training, evaluation, and hyperparameter tuning and cross validations can be found in Appendix C. The code is publicly available.[3]

We report the quality of the learned graph structure (Subsection 3.2) only for the fully connected setting, as the pruned setting trivially has perfect graph quality. The pruned graph only contains the true edges, so the learned graph structure is equal to the true graph structure. For the predictive performance, we use the R2 score to compare the regression models. To average over the datasets, we use the *normalized* R2 score. This normalization is introduced by Wistuba et al. (2015) and used by Feurer et al. (2022); Grinsztajn et al. (2022) for instance. Per dataset, the R2 score is normalized between zero and one, using the worst-performing and the best-performing model on that dataset as the lower and upper bound, respectively.

We run the experiments for three datasets belonging to both dataset types (MVN and SCM), and their graph structures are shown in Appendix B. We compare all explicit GTDL methods that have publicly published code. That is, we compare FiGNN (Li et al., 2019a), T2G-Former (Yan et al., 2023) and INCE (Villaizán-Vallelado et al., 2024). The remainder of the explicit GTDL methods, DRSA-Net (Zheng et al., 2023), MPCFIN (Ye et al., 2024) and Table2Graph (Zhou et al., 2022), have not published their code repositories. For implicit, attention-based methods, we take FT-Transformer (Gorishniy et al., 2021) as an illustrative example. In Appendix A, we discuss how these methods use and interpret the learned weighted adjacency matrix, and how we adapt the implementations to compare them. We use the PGM method BDgraph (Mohammadi & Wit, 2019) as a baseline to understand how well GTDL methods should be able to learn the feature interactions. Finally, we include TabPFN and XGBoost as additional baselines given their overall strong performance, to understand the difficulty of the predictive task.

## 4 Results of structure-aware learning in GTDL

To substantiate our claim that GTDL methods should focus on learning the graph structure, we demonstrate that GTDL methods do not accurately learn the feature interactions and that the predictive performance improves when the graph is pruned to its true edges. Results are aggregated per dataset type, see Appendix D for the results per dataset.

### 4.1 Feature interactions

The ROC AUC of the feature interactions is shown in Figure 4. For all GTDL methods, across both datasets, the ROC AUC is approximately 0.5, which is equal to random chance. There is no difference in the values of the adjacency matrix whether there exists a true edge or not. This shows that GTDL methods do not learn an accurate graph structure. Therefore, the learned feature interactions should not be used for interpretability or explainability. Increasing the number of training samples does not change the ROC AUC, indicating that the poor performance of the GTDL methods is not due to insufficient data. This observation holds when increasing the number of training samples up to $10^5$ samples as discussed in Appendix D.2.

PGMs, which focus on learning the graph structure, can learn the feature interactions, while GTDL methods cannot. The PGM method BDgraph has an ROC AUC very close to one for the MVN datasets. Even for the SCM datasets, which have nonlinear feature interactions, BDgraph can achieve reasonable ROC AUC values. The fact that this PGM method, a non-deep learning method, can learn the feature interactions, while these advanced GTDL methods cannot, suggests that GTDL methods have room for improvement in learning the graph structure.

---

[3]https://github.com/elidub/gtdl_fi

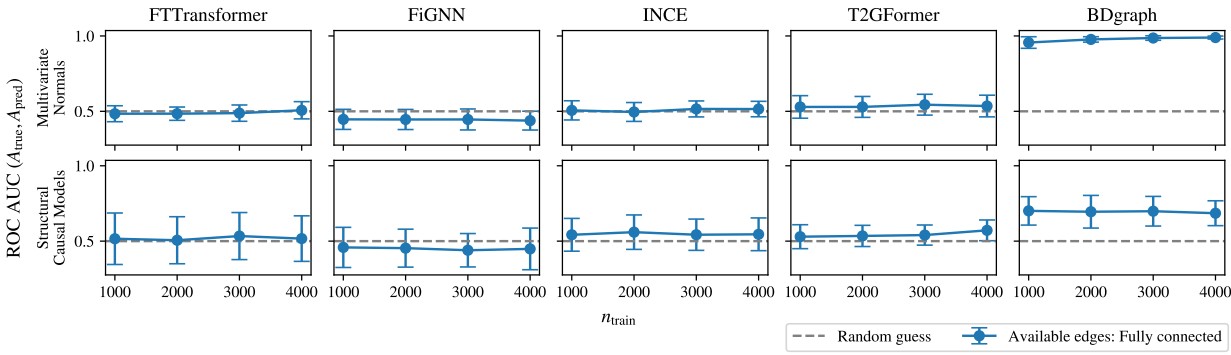

Figure 4: Graph quality in the form of the ROC AUC comparing the learned weighted adjacency matrix with the true binary one, for two different dataset types. All GTDL models have ROC AUC $\approx 0.5$, which is random chance, indicating that they are not able to learn the feature interactions in any meaningful way. The PGM method BDgraph does learn the correct feature interactions better. Error bars show standard deviation across seeds, cross validations and datasets.

## 4.2 Predictive performance

The R2 score of the prediction of the target feature is shown in Figure 5. The key takeaway is that, in general, pruning the graph to the true edges improves the predictive performance of GTDL methods. This result indicates the importance of incorporating accurate structural information into GTDL models. When the graph is pruned to only include true edges, the models are less likely to overfit to spurious or irrelevant feature interactions. Restricting the model to only the true interactions simplifies the optimization landscape, allowing the learning algorithm to focus on meaningful relationships rather than being distracted by false edges. This leads to better generalization and higher predictive accuracy. In contrast, fully-connected models must learn to ignore many false edges, which can introduce noise and make optimization more difficult, especially when data is limited. This finding suggests that the inability of current GTDL methods to recover the true graph structure (as shown by the ROC AUC results) is not just a theoretical issue, but has practical consequences for predictive performance. If the true graph is known or can be estimated reliably, enforcing this structure can provide a boost in performance.

We use the Linear Mixed-Effects Model (Lindstrom & Bates, 1988; Pinheiro & Bates, 2000) to assess whether the pruned graph outperforms the fully connected graph. We elaborate on the statistical testing procedure in Appendix C.1. From Figure 5, we observe that, except for FiGNN, the pruned graph results are both visually (first and third row) and statistically significantly (second and fourth row) better than the fully connected graph results for low number of training samples. In Appendix D.3, we discuss results that indicate that this is because FiGNN treats the task of predicting the target feature as a graph-level task, while all other models treat it as a node-level task.

Regarding baselines, the PGM method BDgraph performs, as expected, well on the MVN datasets, and poorly on the SCM datasets. The SCM datasets have nonlinear feature interactions, while BDgraph can only model linear feature interactions. In most cases, GTDL methods outperform XGBoost, while TabPFN outperforms GTDL methods. A possible explanation for the performance gap is that GTDL and TabPFN may better exploit the tabular structure of data through learned representations, whereas XGBoost relies on an ensemble of weak learners that may not fully capture feature interactions inherent in such structures. This underlines the need for deep learning methods due to their flexibility to learn nonlinear relationships.

Furthermore, the benefit of incorporating the true graph increases by reducing the number of training samples. When ample data is available, models benefit less from incorporating the graph structure correctly, but when data is scarce, leveraging the graph structure improves the predictions. This is in line with the general notion of geometric deep learning, where symmetries in the data are used to improve the learning process (Bronstein et al., 2021). When data is scarce, explicitly incorporating the symmetries from the data into the model is beneficial. However, an abundance of data facilitates an implicit learning of these symmetries (Marchetti

et al., 2024), reducing the benefit of explicitly incorporating them. This observation holds when increasing the number of training samples up to $10^5$ samples, discussed in Appendix D.2.

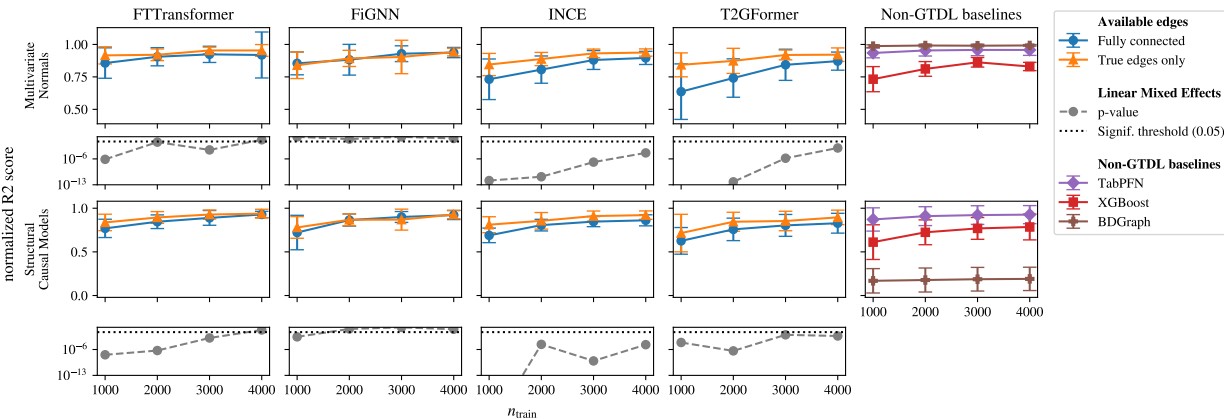

Figure 5: Predictive performance for two different dataset types. When the graph is pruned to its true edges, the predictive performance is, in most cases, better compared to the fully connected graph. The difference reduces as the number of training samples increases. This is confirmed by the *p*-values of the Linear Mixed-Effects Model, where lower *p*-values indicate stronger evidence that the pruned graph results are better than the fully connected graph results. Error bars show standard deviation across seeds, cross validations and datasets.

## 5    Conclusion, limitations, and future work

In this work, we have analyzed the capability of graph-based tabular deep learning (GTDL) methods to learn feature interactions in tabular data. Inspired by the principles of probabilistic graphical models (PGMs), we proposed to use synthetic tabular datasets with known ground-truth graph structures, enabling the GTDL community to quantitatively assess whether models accurately capture the intended graph structure. Current GTDL approaches often produce graph structures used for interpretation, yet our analysis shows that these structures fail to reflect the true interactions among features. This indicates that the mechanisms of message-passing in GNNs, and attention in transformers, does not work as intended for tabular data. Our empirical findings demonstrate that when models operate on accurate interaction structures, predictive performance improves, highlighting that structural fidelity is not merely a matter of explainability, but a core driver of performance.

A potential risk could be the interpretation of the attention map as a weighted adjacency matrix. First, the attention mechanism is not designed to specifically model feature interactions or for explainability. If attention can be used for explainability is an ongoing debate (Bibal et al., 2022; Lopardo et al., 2024), although this debate in the literature mainly focuses around natural language tasks. However, our interpretation for tabular data—that non-interacting features will exhibit low attention values while interacting features will manifest higher attention—is deliberately modest and pragmatic, rather than relying on attention as a fully explanatory tool. Second, while we acknowledge that aggregating attention across layers and heads might obscure certain signals, we find no strong evidence that alternative aggregation strategies would significantly improve graph recovery or alter the overall conclusions.

We highlight three directions how the analysis of this work can be extended in future work. First, future work should move beyond evaluating the learned graph structure (i.e., the presence of edges), but also consider the functional form of the feature interactions (i.e., the type of edges). Learning *how* features interact, and *in what way*, allows for more nuanced, robust, and interpretable modeling of feature relationships. Second, the evaluated datasets and graphs could be more expressive and challenging. Examples include larger graphs with richer topology, missing nodes, and more complex feature interactions involving categorical features, as well as real-world datasets with known ground-truth structure (e.g., knowledge graphs). However, our

results showed that GTDL methods struggle to learn relatively simple, small graph structures, which suggests that improving robustness and structure induction on basic cases remains a priority before scaling to such settings. Finally, structure-aware modeling should be extended beyond flat tables to richer data modalities. Time-series data (Padhi et al., 2021) and relational databases (Fey et al., 2023; Robinson et al., 2024; Cong et al., 2024; Dwivedi et al., 2025) present new challenges for learning and validating feature interactions over time or across relational contexts. Relational deep learning (RDL) (Fey et al., 2023) could ask the same question as we propose GTDL should do: how do columns/features within a table relate? Currently, table-row embeddings in RDL lack structural bias from the graph structure of columns. Furthermore, our approach could be extended from table-level to relational database-level. When doing node-level prediction, does RDL rely on meaningful primary-foreign relationships?

Future work should build on the insights of this work to develop GTDL methods that more effectively learn and leverage feature interactions in tabular data. By prioritizing structural fidelity alongside predictive accuracy, future GTDL models can unlock their full potential.

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

## A  Implementations from literature

This section contains implementation details of the GTDL models that are evaluated in Section 4. In the following Appendices A.1 to A.4, each first paragraph explains what kind of graph is learned, and where and how this is used by the original authors. In the subsections, we discuss the adaptations we made to make the learned graph structures compatible with our discussion and implementation.

**Additional notation.**  We use $N$ for the number of samples, $L$ as the number of layers in the network, $H$ as the number of transformer heads, and $p$ as the number of features. To indicate values that are ranging between zero and one, we denote the corresponding set as $\mathbb{R}_{[0,1]}$.

**Interpreting the attention map as a weighted adjacency matrix.**  For most methods (FT-Transformer, T2G-Former and FiGNN), the learned graph comes from averaging the attention map $a \in \mathbb{R}_{[0,1]}^{N \times L \times H \times p \times p}$ over the samples, layers, and heads. This attention map is normalized with softmax across the last dimension. We note individual values from the attention map as $a_{ilhjk}$, where $i$ is the sample index, $l$ is the layer index, $h$ is the head index, and $j, k$ are the feature indices. So the average attention map is $a_{jk} = \frac{1}{N \times L \times H} \sum_{ilh} a_{ilhjk} \in \mathbb{R}_{[0,1]}^{p \times p}$.

We want to interpret the average attention map $a_{jk}$ as the weighted adjacency matrix $A_{jk}$. For this, we have to 'denormalize' the average attention map. As the attention map $a$ is normalized with a softmax, the last dimension (the rows in the attention map per individual layer and head) sum to one, such that $\sum_k a_{ilhjk} = 1$ for all $i, l, h, j$. This gives a problem, as the maximum value the attention map $a_{ilhjk}$ can have cannot have

two values close to 1 in the same row, while the weighted adjacency matrix $A_{jk}$ should be able to have multiple values close to 1 in the same row.

To 'denormalize' the attention map, we add two steps. First, we set the diagonal of the attention map to zero, as the self-interactions should not be taken into account during evaluation of the feature interactions:

$$a_{ilhjk} = 0 \quad \forall j = k.$$

Second, we divide the attention maps by the maximum value across the row to obtain the adjacency matrices:

$$A_{ilhjk} = a_{ilhjk} / \max_k(a_{ilhjk}).$$

By doing this, all the values with the highest attention across that row now have a value of 1 in the weighted adjacency matrix. Ignoring the diagonal in the attention map is a key step in this procedure: If the model learns that it should not give high attention to the non-diagonal values (as those features are not related), the model should learn to give high attention to the diagonal values. The diagonal values are excluded in the denormalization and do not affect the adjacency matrix.

### A.1 FT-Transformer

FT-Transformer (Gorishniy et al., 2021) (`https://github.com/yandex-research/rtdl-revisiting-models`) learns the attention map $a \in \mathbb{R}_{[0,1]}^{N \times L \times H \times p \times p}$. (Gorishniy et al., 2021) interpret in Section 5.3 the average attention map $a_{jk} = \frac{1}{N \times L \times H} \sum_{ilh} a_{ilhjk} \in \mathbb{R}_{[0,1]}^{p \times p}$ as feature importance. For a few real-world datasets, they compare it to Integrated Gradients (Sundararajan et al., 2017) using rank correlation and find that it performs similarly.

As the attention map is normalized with the softmax, we denormalize it to obtain the weighted adjacency matrix $A$ as described above. This is the only post-hoc adaptation we made to the original implementation.

### A.2 T2G-Former

T2G-Former (Yan et al., 2023) (`https://github.com/jyansir/t2g-former`) learn a feature-relation graph (FR-Graph) $a \in \mathbb{R}_{[0,1]}^{N \times L \times H \times p \times p}$ (Equation 6 in (Yan et al., 2023)). The strength of the graph can be interpreted as the strength of the relations between the features. Section 5.3 and Figure 3 in (Yan et al., 2023) show the FR-Graph for two real-world datasets.

Instead of the Hadamard product in equation 6 in (Yan et al., 2023) to construct the FR-Graph, we use a sum, consistent with their code implementation of the FR-Graph. (Yan et al., 2023) present the FR-Graph from the first and the last layer of the network. We assume that these are averaged over the samples and the heads. Instead, we average over all the layers, following the approach of FT-Transformer. As the FR-Graph is normalized with the softmax, we denormalize the FR-Graph to obtain the weighted adjacency matrix $A$ as described above.

### A.3 INCE

INCE (Villaizán-Vallelado et al., 2024) (`https://github.com/MatteoSalvatori/INCE`) learn edge embeddings $e \in \mathbb{R}^{N \times (p(p-1)) \times d_{\mathrm{emb}}}$ with $d_{\mathrm{emb}}$ the embedding dimension and $(p(p-1))$ the number of edges in a fully connected graph excluding self-loops. Section 6.2 of (Villaizán-Vallelado et al., 2024) presents an algorithm to calculate the feature-feature interaction $p_{\mathrm{int}} \in \mathbb{R}_{[0,1]}^{p \times p}$ from the edge embeddings. Figure 11 in (Villaizán-Vallelado et al., 2024) shows the feature-feature interaction $p_{\mathrm{int}}$ on a real-world dataset.

A lower value of $p_{\mathrm{int}}$ implies more significance. Therefore, we apply one additional step to obtain the weighted adjacency matrix $A = 1 - p_{\mathrm{int}}$.

### A.4 FiGNN

FiGNN (Li et al., 2019a) ([https://github.com/CRIPAC-DIG/Fi_GNN/tree/](https://github.com/CRIPAC-DIG/Fi_GNN/tree/) `7e207b2ffb4f25b63d2079cf7761d09e5dedf6e8`[4]) learn a feature graph in the form of attentional edge weights $a \in \mathbb{R}_{[0,1]}^{N \times (p-1) \times (p-1)}$ for the input features (equations 4 and 5 in Li et al. (2019a)). The edge weights are interpreted as the importance of the interactions. Therefore, they are used to providing explanations on the relationship between different features. In Section 4.5 and Figure 5, the edge weights are presented as a heat map, and are used to explain the relations between features on a real-world dataset.

As the attention edge weights are normalized with the softmax, we denormalize them to obtain the weighted adjacency matrix $A$ as described above. The code implementation of FiGNN is published with TensorFlow. As our implementation is in PyTorch, we have adapted the code to PyTorch. The learned adjacency matrix is of size $(p-1) \times (p-1)$, we impute an additional row and column for the target feature with values of zero.

There are different implementation versions of FiGNN. The first version of FiGNN has been presented at CIKM in November 2019, which is identical to version 1 on ArXiv (Li et al., 2019b). We use the implementation of this version. In July 2020, version 2 on ArXiv (Li et al., 2020) was published, and the main repository ([https://github.com/CRIPAC-DIG/Fi_GNN/](https://github.com/CRIPAC-DIG/Fi_GNN/)) was updated accordingly. Version 2 has some additional attention layers. Furthermore, when inspecting the published code. We observed that this second version does not have a trainable feature graph in its code implementation. Therefore, we stick to the original code implementation of version 1 ([https://github.com/CRIPAC-DIG/Fi_GNN/tree/](https://github.com/CRIPAC-DIG/Fi_GNN/tree/) `7e207b2ffb4f25b63d2079cf7761d09e5dedf6e8`).

## B Data generation

In this section, we describe the graph and data generation process of the two synthetic dataset approaches introduced in Subsection 3.1 and their hyperparameters used in Section 4.

**Multivariate normals.** We follow the default procedure of generating conditional multivariate data, (Mohammadi & Wit, 2015):

(i) Sample a true graph structure $G_{\text{true}} \in \mathbb{R}^{p \times p}$ from the Bernoulli distribution with an edge inclusion probability $P_{\text{edge}}$.

(ii) Sample a covariance matrix $\Sigma_G \in \mathbb{R}^{p \times p}$ from the G-Wishart distribution (Roverato, 2002; Letac & Massam, 2007), which is conditioned on the graph structure $G_{\text{true}}$;[5]

(iii) Obtain $n$ samples $D \in \mathbb{R}^{n \times p}$ from a multivariate normal distribution $\mathcal{N}(0, \Sigma_G)$.

In our experiments, we have $p = 10$ nodes, and an edge inclusion probability of $P_{\text{edge}} = 0.267$. This results in the graph structures as depicted in Figure 6.

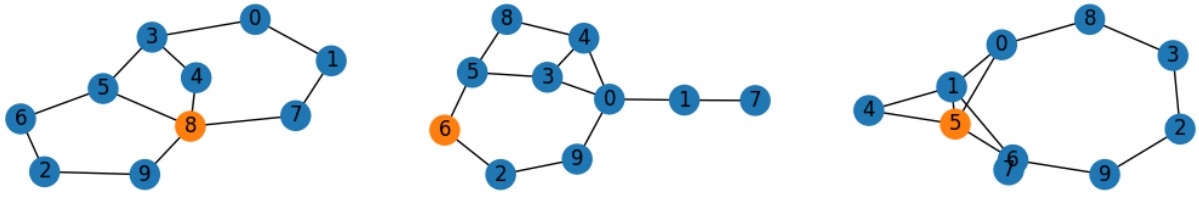

Figure 6: Graphs used in experiments for the MVN1, MVN2 and MVN3 datasets.

---

[4]This on purpose a specific commit, as there are different implementations.

[5]In fact, we sample the precision matrix $K_G = \Sigma_G^{-1}$ form the G-Wishart distribution, and invert it to obtain the covariance matrix $\Sigma_G$. The underscore $\cdot_G$ indicates that the matrix is conditioned on the graph structure $G_{\text{true}}$.

**Structural causal models.** We follow a similar setup as (Hollmann et al., 2025) to generate an SCM and sample data conditional on the graph. They show that with their setup, the synthesized data is similar to real-world tabular data.

(i) Randomly sample a DAG, with $n_{\text{root}}$ root nodes and $p$ child nodes with a probability of $P_{\text{edge}}$ of an incoming edge. The undirected graph structure $G_{\text{true}}$ is obtained by moralizing and marginalizing the DAG (Cowell et al., 1999). Moralizing makes the graph undirected by dropping the direction of the existing edges and connecting all parents of a child node. Marginalizing removes the root nodes from the graph, as they are not part of the dataset $D$. When marginalizing a node, we connect all neighbors of the removed node to each other. In this work, only the root nodes are marginalized. This makes the order of moralization and marginalization irrelevant. The red lines in Figure 7 show the new edges that are added by moralization and marginalization.

(ii) Randomly sample deterministic computational mappings $f_i$ for each child node $i$ in the graph, where the mappings are smooth nonlinear functions, randomly picked from the set of maps listed in Table 2. A computational map defines how a child node $i$ is computed from its parents. They take all the values of the incoming edges as input, and the output is the value of the child node $i$.

(iii) Randomly sample root nodes $x_{\text{root}} \sim \mathcal{N}(0, 1) \in \mathbb{R}^{n \times n_{\text{root}}}$ and traverse the DAG in a topological order, $x_i = f_i(x_{i,\text{parents}}) \in \mathbb{R}^{n \times 1}$. Each output $x_i$ is normalized, clipped between $(-3, 3)$, and Gaussian noise $\mathcal{N}(0, 0.5)$ is added. This is summarized by

$$x_i = \text{clip}(\text{normalize}(f_i(x_{i,\text{parents}})) + \mathcal{N}(0, 0.5), -3, 3). \qquad (2)$$

We consider all the traversed outputs $x_i$ as the dataset $D \in \mathbb{R}^{n \times p}$.

Each DAG has $p = 10$ nodes. These $p$ nodes are evenly distributed over $n_{\text{DAG layers}} = 3$ layers, where each layer has a minimum of 3 nodes. The DAG has a 'zeroth' layer of $n_{\text{root}} = 3$ root nodes. This means that each layer has 3 or 4 nodes. Each node has a $P_{\text{edge}} = 0.5$ probability of having an edge to the nodes in the next layer. With these hyperparameters, the three DAGs that are used in Section 4 are shown in Figure 7.

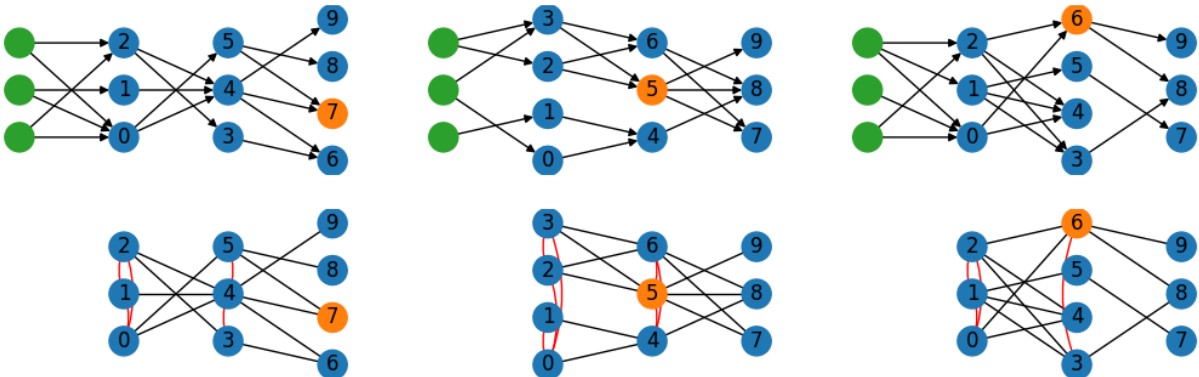

Figure 7: Top: DAGs used in experiments for the SCM1, SCM2 and SCM3 datasets. Bottom: Corresponding undirected graph structures $G_{\text{true}}$ after moralization and marginalization. Moralized and marginalized edges are depicted in red.

Table 2: Computational maps

| # parents | $f(x_{\text{parents}})$ |
|---|---|
| 1 | $x_1^2/3$ |
| | $0{,}5\,x_1^2 + 3\,x_1$ |
| | $-|x_1| + 4\,x_1$ |
| 2 | $(x_1 x_2 + x_1^2)/2$ |
| | $x_1^2 + x_2^2 - x_1 x_2$ |
| | $-(x_1 + x_2)^2 + x_1 x_2$ |
| 3 | $(x_1 x_2 + x_3^2)/3$ |
| | $-x_1^2 + x_2 x_3 + x_3$ |
| | $(x_1 + x_2 + x_3) + x_1 x_3$ |

## C  Experiment details

**Data splitting.**  We adapt our train, validation, and test splitting and our tuning strategy to balance between a fair comparison between different dataset sizes and an efficient hyperparameter tuning.

Following (Grinsztajn et al., 2022), we differentiate between a validation set used for early stopping $D_{\text{val, early stop}}$ and a validation set used for hyperparameter tuning $D_{\text{val, hparam}}$, such that we have four disjoint sets: $D_{\text{train}}$, $D_{\text{val, early stop}}$, $D_{\text{val, hparam}}$, and $D_{\text{test}}$. We vary the number of training samples $n_{\text{train}}$ in our experiments between 1000 and 4000, and set both $n_{\text{test}} = n_{\text{val, hparam}} = 2500$ and $n_{\text{val, early stop}} = 0.25 n_{\text{train}}$.

In our experiments, we do not change $D_{\text{val, hparam}}$ and $D_{\text{test}}$ to limit the number of cross-validation and iterations we have to do. We randomly sample $D_{\text{train}}$ and $D_{\text{val, early stop}}$ for each fold. This strategy is visualized in Figure 8. For $n_{\text{train}} = 1000$ samples we evaluate over 4 folds, $n_{\text{train}} = 2000$ over 3 fold, for $n_{\text{train}} = 3000$ over 2 folds and for $n_{\text{train}} = 4000$ over 1 fold.

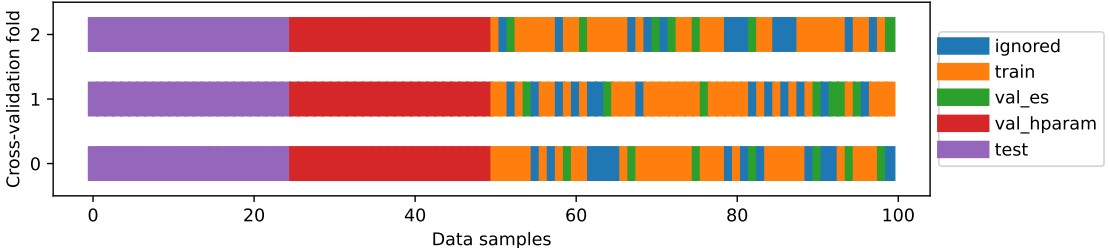

Figure 8: Splitting strategy for the example $N = 100$ and $n_{\text{train}} = 0.3N$. The training set and the validation set for early stopping are randomly sampled for each fold. The test set and validation set for hyperparameter tuning are fixed.

**Training and evaluation.**  We minimize the MSE loss function and optimize using Adam (Kingma & Ba, 2017) with a fixed batch size of 256 and tune the learning rate together with the other model hyperparameters. We continue training until the validation loss does not improve for 10 epochs. There is a theoretical upper bound of 400 epochs, which is rarely reached in practice. We select the best hyperparameters that minimize the MSE on the separate hyperparameter validation set. After tuning, we run 10 runs per cross-validation fold. We report the R2 score to evaluate the predictive performance of the target feature, and the ROC AUC to evaluate the learned feature interactions.

**Hyperparameter tuning.**  For every combination of network, dataset, and $n_{\text{train}}$, we tune the model's hyperparameters and the learning rate. For all models, we use tree-structured Parzen estimator (TPE) (Bergstra et al., 2011), a Bayesian optimization technique within the Optuna library (Akiba et al., 2019). We run a total of 50 trials for each setting, where the first trial has the default hyperparameters of the implementation. We keep the default setting of the Optuna implementation, where the first 10 trials are

done with random search. The hyperparameters of TabPFN are not tuned, as it is a pretrained model that can get accurate predictions without fine-tuning (Hollmann et al., 2025).

The hyperparameter distribution and the default hyperparameters of all models are listed in Tables 3 to 7. For all models, the search space and default values are taken from the original implementations if not specified otherwise in the caption of the tables. The search space of layer count and embedding size is set the same for fairer comparison across models. The distribution space of the learning rate is $\text{LogUniform}[10^{-5}, 10^{-3}]$ with a default value of $10^{-3}$ for all models.

Table 3: FT-Transformer (Gorishniy et al., 2021) hyperparameter space.

| Parameter | Distribution | Default |
|---|---|---|
| Layer count | $\text{UniformInt}[1, 6]$ | 3 |
| Embedding size | $\{8, 16, 32, 64, 128, 264\}$ | 128 |
| Attention head count | - | 8 |
| Attention dropout | $\text{Uniform}[0.0, 0.5]$ | 0.2 |
| FFN size factor | $\text{Uniform}[^2/_3, ^7/_3]$ | $^4/_3$ |
| FFN dropout | $\text{Uniform}[0.0, 0.5]$ | 0.1 |
| Residual dropout | $\text{Uniform}[0.0, 0.2]$ | 0 |

Table 4: T2G-Former (Yan et al., 2023) hyperparameter space. Default values are taken from the same as from FT-Transformer.

| Parameter | Distribution | Default |
|---|---|---|
| Layer count | $\text{UniformInt}[1, 6]$ | 3 |
| Embedding size | $\{8, 16, 32, 64, 128, 264\}$ | 128 |
| Attention head count | - | 8 |
| Attention dropout | $\text{Uniform}[0.0, 0.5]$ | 0.2 |
| FFN size factor | $\text{Uniform}[^2/_3, ^7/_3]$ | $^4/_3$ |
| FFN dropout | $\text{Uniform}[0.0, 0.5]$ | 0.1 |

Table 5: INCE (Villaizán-Vallelado et al., 2024) hyperparameter space.

| Parameter | Distribution | Default |
|---|---|---|
| Layer count | $\text{UniformInt}[1, 6]$ | 4 |
| Embedding size | $\{8, 16, 32, 64, 128, 264\}$ | 128 |
| MLP layer count | $\{1, 2, 3, 4\}$ | 3 |
| Dropout | $\text{Uniform}[0.0, 0.5]$ | 0 |

Table 6: FiGNN (Li et al., 2019a) hyperparameter space. The distribution space was not shared by the original implementation.

| Parameter | Distribution | Default |
|---|---|---|
| Layer count | $\text{UniformInt}[1, 6]$ | 3 |
| Embedding size | $\{8, 16, 32, 64, 128, 264\}$ | 16 |
| Dropout | $\text{Uniform}[0.0, 0.5]$ | 0 |

Table 7: XGBoost (Chen & Guestrin, 2016) hyperparameter space. Default values are taken from the official implementation (Chen et al., 2026), distribution space is the same as in (Grinsztajn et al., 2022).

| Parameter | Distribution | Default |
|---|---|---|
| Max depth | UniformInt$[3, 10]$ | 6 |
| Min child weight | LogUniform$[10^{-8}, 10^5]$ | 1 |
| Subsample | Uniform$[0.5, 1.0]$ | 1.0 |
| Learning rate (eta) | LogUniform$[10^{-5}, 1.0]$ | 0.3 |
| Colsample by level | Uniform$[0.5, 1.0]$ | 1.0 |
| Colsample by tree | Uniform$[0.5, 1.0]$ | 1.0 |
| Gamma | LogUniform$[10^{-8}, 100]$ | $10^{-8}$ |
| Lambda | LogUniform$[10^{-8}, 100]$ | 1.0 |
| Alpha | LogUniform$[10^{-8}, 100]$ | $10^{-8}$ |

### C.1 Statistical testing

To assess the statistical significance of the predictive performance improvement when using the pruned graph compared to the fully connected graph, we used a Linear Mixed Effects Model (Lindstrom & Bates, 1988; Pinheiro & Bates, 2000). We selected this approach over aggregating results per fold (e.g., for a Wilcoxon signed-rank test (Wilcoxon, 1945)) to avoid the loss of statistical power, given the limited number of cross-validation folds $(4, 3, 2, 1)$ available for $n_{\text{train}} = (1000, 2000, 3000, 4000)$, respectively. Since we evaluate 10 random seeds per fold, the data exhibits a hierarchical structure. Treating these seeds as independent samples would violate the independence assumption of standard tests and lead to pseudoreplication (Nadeau & Bengio, 1999). Therefore, we modeled the cross-validation fold as a random effect to account for the correlation between seeds, and the graph type (pruned or fully connected) as a fixed effect. Although not documented in this work, we compared the Linear Mixed Effects Model with the one-sided Mann-Whitney U test (Mann & Whitney, 1947) and found comparable $p$-values.

## D   Additional results

### D.1 Results per dataset

In Section 4 we have discussed the results of the learned graph structure and the predictive performance of the target feature while aggregating over three datasets per the dataset type MVN and SCM. In Figure 9 and Table 8 we show the results per individual dataset for the learned graph structure, in Figure 10 and Table 9 we show them for the predictive performance. The results are consistent with the results shown in Figure 4 and Figure 5; no new insights are gained.

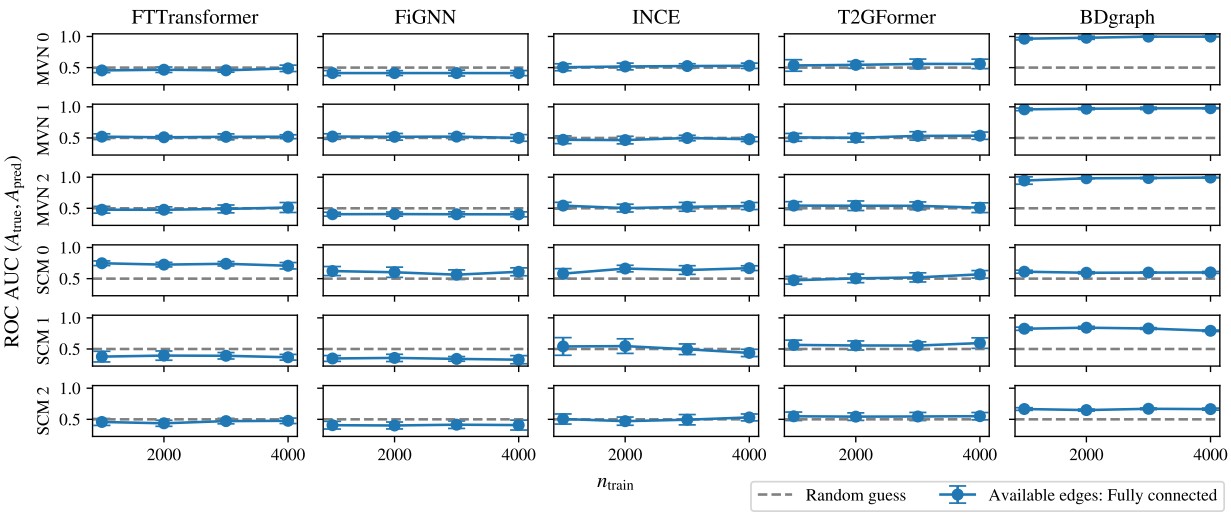

Figure 9: Graph quality in the form of the ROC AUC comparing the learned weighted adjacency matrix with the true binary one, for six different datasets. Most GTDL models have ROC AUC $\approx 0.5$, which is random chance, indicating that they are not able to learn the feature interactions in any meaningful way. The PGM method BDgraph can learn the feature interactions. Error bars show standard deviation across seeds and cross validations. See Figure 4 for the results aggregated over the two dataset types.

Table 8: Graph quality in the form of the ROC AUC comparing the learned weighted adjacency matrix with the true binary one, for six different datasets and five different models. The statistics $\mu$ and $\sigma$ represent the mean and standard deviation aggregated over seeds and cross validations. See Figure 9 for the same results.

| | Model | FTTransformer | | FiGNN | | INCE | | T2GFormer | | BDgraph | |
| | Graph | Fully connected | | Fully connected | | Fully connected | | Fully connected | | Fully connected | |
| | Statistic | $\mu$ | $\sigma$ | $\mu$ | $\sigma$ | $\mu$ | $\sigma$ | $\mu$ | $\sigma$ | $\mu$ | $\sigma$ |
| Dataset | $n_{\text{train}}$ | | | | | | | | | | |
| MVN 0 | 1000 | 4.548e-01 | 3.508e-02 | 4.121e-01 | 4.071e-02 | 5.057e-01 | 5.581e-02 | 5.344e-01 | 9.255e-02 | 9.639e-01 | 1.910e-02 |
| | 2000 | 4.658e-01 | 4.352e-02 | 4.121e-01 | 3.872e-02 | 5.179e-01 | 5.411e-02 | 5.441e-01 | 5.511e-02 | 9.790e-01 | 2.515e-02 |
| | 3000 | 4.560e-01 | 2.904e-02 | 4.122e-01 | 4.682e-02 | 5.249e-01 | 3.699e-02 | 5.592e-01 | 7.778e-02 | 9.976e-01 | 1.733e-03 |
| | 4000 | 4.883e-01 | 5.100e-02 | 4.116e-01 | 4.091e-02 | 5.295e-01 | 4.251e-02 | 5.595e-01 | 7.586e-02 | 9.975e-01 | 0.000e+00 |
| MVN 1 | 1000 | 5.203e-01 | 4.563e-02 | 5.226e-01 | 4.434e-02 | 4.704e-01 | 6.147e-02 | 5.096e-01 | 6.267e-02 | 9.594e-01 | 2.092e-02 |
| | 2000 | 5.108e-01 | 2.726e-02 | 5.184e-01 | 5.335e-02 | 4.658e-01 | 6.090e-02 | 5.025e-01 | 6.883e-02 | 9.697e-01 | 1.518e-02 |
| | 3000 | 5.178e-01 | 4.747e-02 | 5.216e-01 | 4.772e-02 | 4.989e-01 | 4.354e-02 | 5.327e-01 | 6.719e-02 | 9.753e-01 | 1.896e-02 |
| | 4000 | 5.200e-01 | 3.107e-02 | 5.003e-01 | 5.355e-02 | 4.793e-01 | 3.654e-02 | 5.360e-01 | 5.847e-02 | 9.773e-01 | 7.434e-03 |
| MVN 2 | 1000 | 4.768e-01 | 5.439e-02 | 4.049e-01 | 3.134e-02 | 5.429e-01 | 5.289e-02 | 5.431e-01 | 6.307e-02 | 9.460e-01 | 5.934e-02 |
| | 2000 | 4.763e-01 | 4.671e-02 | 4.069e-01 | 3.329e-02 | 5.032e-01 | 6.248e-02 | 5.410e-01 | 7.662e-02 | 9.822e-01 | 5.646e-03 |
| | 3000 | 4.899e-01 | 6.302e-02 | 4.041e-01 | 3.894e-02 | 5.236e-01 | 7.065e-02 | 5.398e-01 | 6.247e-02 | 9.861e-01 | 8.534e-03 |
| | 4000 | 5.127e-01 | 7.981e-02 | 4.029e-01 | 3.976e-02 | 5.364e-01 | 5.616e-02 | 5.097e-01 | 7.822e-02 | 9.934e-01 | 2.715e-03 |
| SCM 0 | 1000 | 7.471e-01 | 3.716e-02 | 6.218e-01 | 7.268e-02 | 5.805e-01 | 8.022e-02 | 4.739e-01 | 6.207e-02 | 6.113e-01 | 2.409e-02 |
| | 2000 | 7.250e-01 | 3.665e-02 | 6.024e-01 | 8.207e-02 | 6.618e-01 | 5.272e-02 | 5.038e-01 | 6.840e-02 | 5.945e-01 | 1.699e-02 |
| | 3000 | 7.387e-01 | 3.545e-02 | 5.639e-01 | 7.701e-02 | 6.381e-01 | 6.953e-02 | 5.201e-01 | 7.332e-02 | 5.973e-01 | 8.906e-03 |
| | 4000 | 7.071e-01 | 5.140e-02 | 6.099e-01 | 6.310e-02 | 6.683e-01 | 3.808e-02 | 5.694e-01 | 6.003e-02 | 5.996e-01 | 1.497e-02 |
| SCM 1 | 1000 | 3.775e-01 | 8.564e-02 | 3.476e-01 | 4.642e-02 | 5.406e-01 | 1.419e-01 | 5.655e-01 | 7.674e-02 | 8.255e-01 | 2.569e-02 |
| | 2000 | 3.938e-01 | 7.435e-02 | 3.567e-01 | 5.796e-02 | 5.463e-01 | 1.173e-01 | 5.564e-01 | 7.292e-02 | 8.420e-01 | 2.206e-02 |
| | 3000 | 3.908e-01 | 5.154e-02 | 3.409e-01 | 3.738e-02 | 4.957e-01 | 8.539e-02 | 5.550e-01 | 5.998e-02 | 8.288e-01 | 1.437e-02 |
| | 4000 | 3.671e-01 | 4.565e-02 | 3.280e-01 | 6.537e-02 | 4.369e-01 | 5.861e-02 | 5.946e-01 | 8.459e-02 | 7.921e-01 | 1.123e-02 |
| SCM 2 | 1000 | 4.575e-01 | 5.500e-02 | 4.058e-01 | 6.178e-02 | 5.050e-01 | 8.119e-02 | 5.489e-01 | 6.800e-02 | 6.656e-01 | 2.108e-02 |
| | 2000 | 4.364e-01 | 4.861e-02 | 4.013e-01 | 5.555e-02 | 4.706e-01 | 6.578e-02 | 5.433e-01 | 6.028e-02 | 6.486e-01 | 1.562e-02 |
| | 3000 | 4.714e-01 | 4.357e-02 | 4.139e-01 | 6.014e-02 | 4.938e-01 | 8.293e-02 | 5.456e-01 | 6.480e-02 | 6.692e-01 | 1.296e-02 |
| | 4000 | 4.762e-01 | 4.352e-02 | 4.077e-01 | 7.953e-02 | 5.308e-01 | 5.504e-02 | 5.507e-01 | 5.860e-02 | 6.641e-01 | 1.610e-02 |

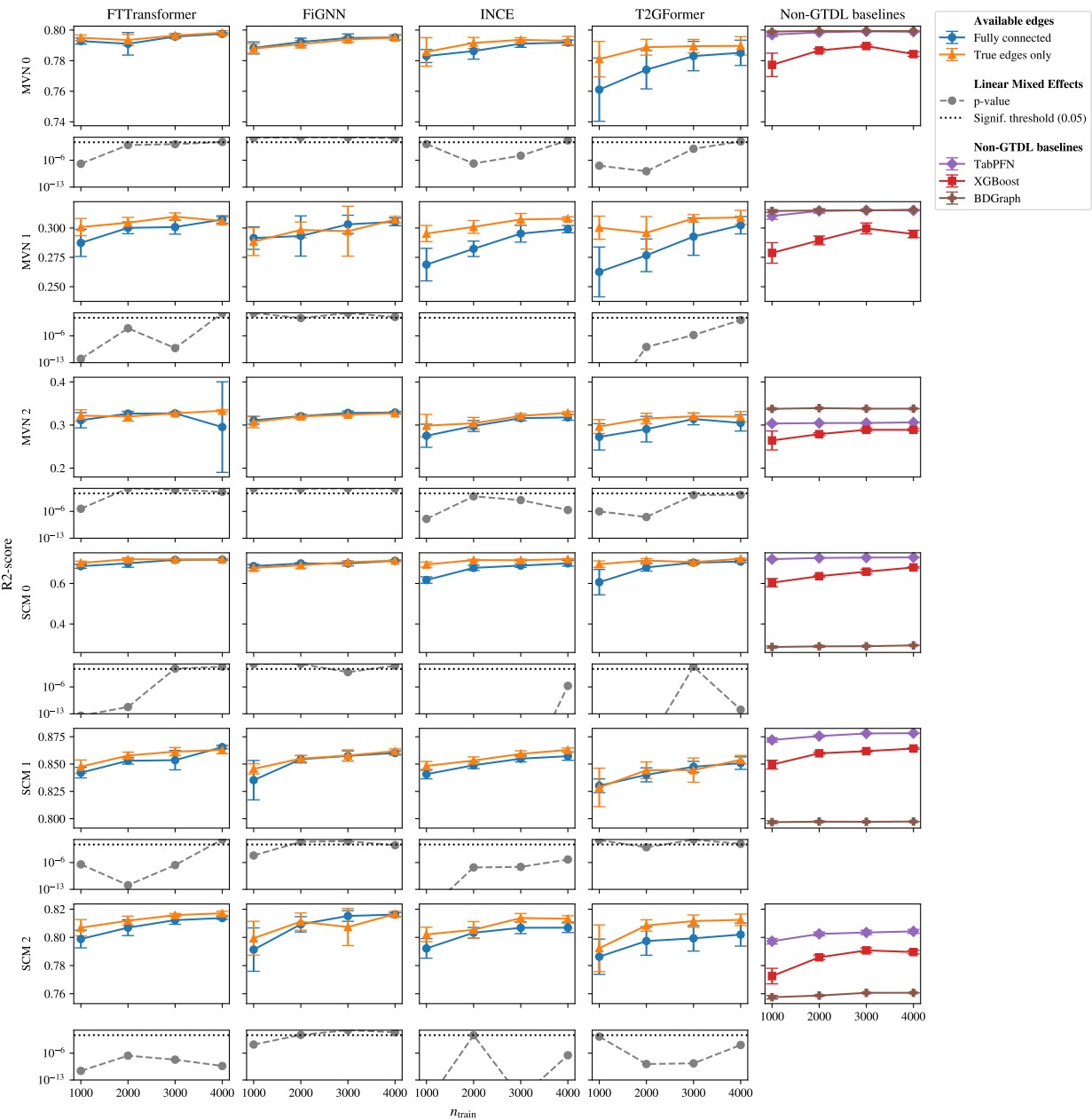

Figure 10: Predictive performance for six different datasets. When the graph is pruned to its true edges, the predictive performance is, in most cases, better compared to the fully connected graph. The difference reduces as the number of training samples increases. This is confirmed by the $p$-values of the Linear Mixed-Effects Model, where lower $p$-values indicate stronger evidence that the pruned graph results are better than the fully connected graph results. When $p$-values are not visible, they are below the lower limit of $10^{-13}$. Error bars show standard deviation across seeds and cross validations. See Figure 5 for the results aggregated over the two dataset types.

Table 9: Predictive performance in the form of $R^2$, for six different datasets and five different models. We compare the performance when the graph is fully connected versus when it is pruned to its true edges only. The statistics $\mu$ and $\sigma$ represent the mean and standard deviation aggregated over seeds and cross validations. With the Linear Mixed Effects (LME) Model we test if the $R^2$ results obtained with the pruned graph are greater than those with the fully connected graph. See Figure 10 for the same results.

| Dataset | Model | Graph Statistic $n_{\text{train}}$ | Fully connected $\mu$ | $\sigma$ | True edges only $\mu$ | $\sigma$ | LME $p$-value |
|---|---|---|---|---|---|---|---|
| MVN 0 | FTTransformer | 1000 | 7.928e-01 | 2.000e-03 | 7.950e-01 | 1.782e-03 | 1.221e-07 |
| | | 2000 | 7.910e-01 | 7.379e-03 | 7.935e-01 | 3.681e-03 | 9.239e-03 |
| | | 3000 | 7.957e-01 | 1.114e-03 | 7.965e-01 | 1.374e-03 | 1.568e-02 |
| | | 4000 | 7.975e-01 | 7.178e-04 | 7.983e-01 | 1.299e-03 | 5.861e-02 |
| | FiGNN | 1000 | 7.883e-01 | 3.955e-03 | 7.879e-01 | 3.121e-03 | 6.958e-01 |
| | | 2000 | 7.922e-01 | 2.499e-03 | 7.908e-01 | 2.436e-03 | 9.934e-01 |
| | | 3000 | 7.948e-01 | 2.658e-03 | 7.938e-01 | 1.819e-03 | 9.122e-01 |
| | | 4000 | 7.952e-01 | 1.693e-03 | 7.951e-01 | 1.575e-03 | 5.715e-01 |
| | INCE | 1000 | 7.830e-01 | 4.252e-03 | 7.857e-01 | 9.369e-03 | 1.698e-02 |
| | | 2000 | 7.863e-01 | 5.282e-03 | 7.916e-01 | 3.636e-03 | 1.403e-07 |
| | | 3000 | 7.911e-01 | 2.362e-03 | 7.936e-01 | 1.393e-03 | 1.593e-05 |
| | | 4000 | 7.918e-01 | 1.664e-03 | 7.929e-01 | 2.931e-03 | 1.519e-01 |
| | T2GFormer | 1000 | 7.611e-01 | 2.067e-02 | 7.810e-01 | 1.160e-02 | 3.768e-08 |
| | | 2000 | 7.741e-01 | 1.252e-02 | 7.888e-01 | 5.172e-03 | 1.383e-09 |
| | | 3000 | 7.830e-01 | 9.613e-03 | 7.894e-01 | 4.455e-03 | 9.891e-04 |
| | | 4000 | 7.851e-01 | 8.265e-03 | 7.897e-01 | 5.989e-03 | 8.067e-02 |
| | TabPFN | 1000 | 7.970e-01 | 1.230e-03 | | | |
| | | 2000 | 7.985e-01 | 2.419e-04 | | | |
| | | 3000 | 7.991e-01 | 1.497e-04 | | | |
| | | 4000 | 7.989e-01 | 6.862e-05 | | | |
| | XGBoost | 1000 | 7.772e-01 | 7.654e-03 | | | |
| | | 2000 | 7.867e-01 | 1.211e-03 | | | |
| | | 3000 | 7.896e-01 | 9.442e-04 | | | |
| | | 4000 | 7.843e-01 | 1.433e-03 | | | |
| | BDgraph | 1000 | 7.990e-01 | 2.730e-04 | | | |
| | | 2000 | 7.994e-01 | 9.546e-05 | | | |
| | | 3000 | 7.993e-01 | 3.010e-04 | | | |
| | | 4000 | 7.994e-01 | 6.052e-06 | | | |
| MVN 1 | FTTransformer | 1000 | 2.873e-01 | 1.157e-02 | 3.008e-01 | 7.322e-03 | 1.013e-12 |
| | | 2000 | 3.001e-01 | 4.982e-03 | 3.045e-01 | 4.561e-03 | 9.286e-05 |
| | | 3000 | 3.008e-01 | 6.024e-03 | 3.097e-01 | 3.192e-03 | 6.228e-10 |
| | | 4000 | 3.072e-01 | 3.170e-03 | 3.059e-01 | 3.257e-03 | 8.191e-01 |
| | FiGNN | 1000 | 2.914e-01 | 9.587e-03 | 2.884e-01 | 1.186e-02 | 8.939e-01 |
| | | 2000 | 2.932e-01 | 1.712e-02 | 2.984e-01 | 6.712e-03 | 4.367e-02 |
| | | 3000 | 3.031e-01 | 7.763e-03 | 2.973e-01 | 2.127e-02 | 8.810e-01 |
| | | 4000 | 3.053e-01 | 3.281e-03 | 3.071e-01 | 2.695e-03 | 8.243e-02 |
| | INCE | 1000 | 2.688e-01 | 1.387e-02 | 2.953e-01 | 6.880e-03 | 1.989e-35 |
| | | 2000 | 2.823e-01 | 6.577e-03 | 3.009e-01 | 5.433e-03 | 1.105e-43 |
| | | 3000 | 2.952e-01 | 7.278e-03 | 3.073e-01 | 5.125e-03 | 2.980e-17 |
| | | 4000 | 2.991e-01 | 3.093e-03 | 3.080e-01 | 1.606e-03 | 2.957e-16 |
| | T2GFormer | 1000 | 2.626e-01 | 2.122e-02 | 3.002e-01 | 9.836e-03 | 2.980e-26 |
| | | 2000 | 2.767e-01 | 1.393e-02 | 2.959e-01 | 1.383e-02 | 1.249e-09 |

Table 9: Predictive performance in the form of $R^2$, for six different datasets and five different models. We compare the performance when the graph is fully connected versus when it is pruned to its true edges only. The statistics $\mu$ and $\sigma$ represent the mean and standard deviation aggregated over seeds and cross validations. With the Linear Mixed Effects (LME) Model we test if the $R^2$ results obtained with the pruned graph are greater than those with the fully connected graph. See Figure 10 for the same results.

| | | Graph Statistic | Fully connected | | True edges only | | LME |
| | | | $\mu$ | $\sigma$ | $\mu$ | $\sigma$ | $p$-value |
| Dataset | Model | $n_{\text{train}}$ | | | | | |
|---|---|---|---|---|---|---|---|
| | | 3000 | 2.927e-01 | 1.595e-02 | 3.082e-01 | 3.218e-03 | 1.576e-06 |
| | | 4000 | 3.023e-01 | 7.366e-03 | 3.090e-01 | 5.987e-03 | 1.329e-02 |
| | TabPFN | 1000 | 3.103e-01 | 2.905e-03 | | | |
| | | 2000 | 3.146e-01 | 1.909e-03 | | | |
| | | 3000 | 3.151e-01 | 3.399e-04 | | | |
| | | 4000 | 3.149e-01 | 2.604e-04 | | | |
| | XGBoost | 1000 | 2.788e-01 | 8.729e-03 | | | |
| | | 2000 | 2.894e-01 | 3.750e-03 | | | |
| | | 3000 | 2.996e-01 | 4.644e-03 | | | |
| | | 4000 | 2.949e-01 | 3.129e-03 | | | |
| | BDgraph | 1000 | 3.145e-01 | 1.158e-03 | | | |
| | | 2000 | 3.150e-01 | 1.218e-03 | | | |
| | | 3000 | 3.151e-01 | 8.791e-04 | | | |
| | | 4000 | 3.155e-01 | 2.023e-05 | | | |
| MVN 2 | FTTransformer | 1000 | 3.111e-01 | 1.771e-02 | 3.218e-01 | 1.356e-02 | 4.767e-06 |
| | | 2000 | 3.265e-01 | 5.374e-03 | 3.196e-01 | 1.063e-02 | 9.999e-01 |
| | | 3000 | 3.271e-01 | 2.848e-03 | 3.273e-01 | 3.762e-03 | 4.482e-01 |
| | | 4000 | 2.952e-01 | 1.051e-01 | 3.331e-01 | 2.340e-03 | 1.272e-01 |
| | FiGNN | 1000 | 3.110e-01 | 8.880e-03 | 3.069e-01 | 1.322e-02 | 9.635e-01 |
| | | 2000 | 3.207e-01 | 5.036e-03 | 3.198e-01 | 4.937e-03 | 7.556e-01 |
| | | 3000 | 3.280e-01 | 4.337e-03 | 3.240e-01 | 3.670e-03 | 9.993e-01 |
| | | 4000 | 3.290e-01 | 2.607e-03 | 3.273e-01 | 2.265e-03 | 9.404e-01 |
| | INCE | 1000 | 2.751e-01 | 2.681e-02 | 2.985e-01 | 2.614e-02 | 1.103e-08 |
| | | 2000 | 2.977e-01 | 1.265e-02 | 3.040e-01 | 1.372e-02 | 8.489e-03 |
| | | 3000 | 3.159e-01 | 5.979e-03 | 3.211e-01 | 5.549e-03 | 8.908e-04 |
| | | 4000 | 3.177e-01 | 6.735e-03 | 3.283e-01 | 2.787e-03 | 2.291e-06 |
| | T2GFormer | 1000 | 2.725e-01 | 3.057e-02 | 2.964e-01 | 1.609e-02 | 9.802e-07 |
| | | 2000 | 2.904e-01 | 2.975e-02 | 3.150e-01 | 1.239e-02 | 3.412e-08 |
| | | 3000 | 3.140e-01 | 1.346e-02 | 3.203e-01 | 7.734e-03 | 1.832e-02 |
| | | 4000 | 3.049e-01 | 1.888e-02 | 3.193e-01 | 1.188e-02 | 2.071e-02 |
| | TabPFN | 1000 | 3.035e-01 | 1.660e-03 | | | |
| | | 2000 | 3.046e-01 | 5.855e-04 | | | |
| | | 3000 | 3.048e-01 | 3.652e-04 | | | |
| | | 4000 | 3.061e-01 | 3.168e-04 | | | |
| | XGBoost | 1000 | 2.641e-01 | 2.197e-02 | | | |
| | | 2000 | 2.789e-01 | 3.089e-03 | | | |
| | | 3000 | 2.890e-01 | 3.739e-03 | | | |
| | | 4000 | 2.889e-01 | 2.763e-03 | | | |
| | BDgraph | 1000 | 3.376e-01 | 1.335e-03 | | | |
| | | 2000 | 3.392e-01 | 2.941e-04 | | | |
| | | 3000 | 3.380e-01 | 1.755e-04 | | | |
| | | 4000 | 3.380e-01 | 1.424e-05 | | | |
| SCM 0 | FTTransformer | 1000 | 6.852e-01 | 1.038e-02 | 7.012e-01 | 8.936e-03 | 3.191e-14 |

Table 9: Predictive performance in the form of $R^2$, for six different datasets and five different models. We compare the performance when the graph is fully connected versus when it is pruned to its true edges only. The statistics $\mu$ and $\sigma$ represent the mean and standard deviation aggregated over seeds and cross validations. With the Linear Mixed Effects (LME) Model we test if the $R^2$ results obtained with the pruned graph are greater than those with the fully connected graph. See Figure 10 for the same results.

| Dataset | Model | Graph Statistic $n_{\text{train}}$ | Fully connected $\mu$ | $\sigma$ | True edges only $\mu$ | $\sigma$ | LME $p$-value |
|---|---|---|---|---|---|---|---|
| | | 2000 | 6.996e-01 | 1.954e-02 | 7.195e-01 | 7.664e-03 | 5.864e-12 |
| | | 3000 | 7.166e-01 | 5.022e-03 | 7.186e-01 | 3.711e-03 | 6.297e-02 |
| | | 4000 | 7.176e-01 | 4.316e-03 | 7.191e-01 | 3.196e-03 | 1.897e-01 |
| | FiGNN | 1000 | 6.857e-01 | 8.771e-03 | 6.770e-01 | 1.356e-02 | 9.999e-01 |
| | | 2000 | 6.987e-01 | 5.186e-03 | 6.896e-01 | 1.378e-02 | 9.998e-01 |
| | | 3000 | 6.985e-01 | 1.175e-02 | 7.051e-01 | 4.950e-03 | 7.274e-03 |
| | | 4000 | 7.122e-01 | 3.105e-03 | 7.129e-01 | 4.739e-03 | 3.559e-01 |
| | INCE | 1000 | 6.169e-01 | 1.583e-02 | 6.934e-01 | 1.276e-02 | 7.188e-139 |
| | | 2000 | 6.769e-01 | 1.258e-02 | 7.155e-01 | 3.774e-03 | 8.700e-70 |
| | | 3000 | 6.881e-01 | 8.860e-03 | 7.149e-01 | 5.249e-03 | 4.991e-32 |
| | | 4000 | 6.993e-01 | 1.378e-02 | 7.200e-01 | 3.107e-03 | 1.913e-06 |
| | T2GFormer | 1000 | 6.064e-01 | 6.247e-02 | 6.952e-01 | 1.588e-02 | 1.506e-18 |
| | | 2000 | 6.792e-01 | 1.816e-02 | 7.134e-01 | 9.392e-03 | 4.987e-22 |
| | | 3000 | 7.027e-01 | 9.075e-03 | 7.054e-01 | 1.035e-02 | 1.934e-01 |
| | | 4000 | 7.079e-01 | 5.211e-03 | 7.218e-01 | 3.525e-03 | 1.184e-12 |
| | TabPFN | 1000 | 7.199e-01 | 3.512e-03 | | | |
| | | 2000 | 7.260e-01 | 1.418e-03 | | | |
| | | 3000 | 7.278e-01 | 4.063e-04 | | | |
| | | 4000 | 7.288e-01 | 1.264e-04 | | | |
| | XGBoost | 1000 | 6.038e-01 | 2.008e-02 | | | |
| | | 2000 | 6.360e-01 | 4.754e-03 | | | |
| | | 3000 | 6.582e-01 | 1.142e-02 | | | |
| | | 4000 | 6.790e-01 | 3.240e-03 | | | |
| | BDgraph | 1000 | 2.865e-01 | 4.755e-03 | | | |
| | | 2000 | 2.895e-01 | 4.179e-03 | | | |
| | | 3000 | 2.903e-01 | 1.251e-03 | | | |
| | | 4000 | 2.944e-01 | 9.584e-05 | | | |
| SCM 1 | FTTransformer | 1000 | 8.421e-01 | 4.800e-03 | 8.476e-01 | 6.136e-03 | 3.358e-07 |
| | | 2000 | 8.530e-01 | 3.025e-03 | 8.578e-01 | 3.103e-03 | 1.220e-12 |
| | | 3000 | 8.536e-01 | 8.851e-03 | 8.614e-01 | 3.758e-03 | 2.128e-07 |
| | | 4000 | 8.655e-01 | 1.603e-03 | 8.629e-01 | 3.397e-03 | 9.864e-01 |
| | FiGNN | 1000 | 8.353e-01 | 1.802e-02 | 8.456e-01 | 4.649e-03 | 6.706e-05 |
| | | 2000 | 8.544e-01 | 3.347e-03 | 8.549e-01 | 2.867e-03 | 2.557e-01 |
| | | 3000 | 8.575e-01 | 4.599e-03 | 8.579e-01 | 5.205e-03 | 3.690e-01 |
| | | 4000 | 8.601e-01 | 1.891e-03 | 8.618e-01 | 2.076e-03 | 3.270e-02 |
| | INCE | 1000 | 8.407e-01 | 4.241e-03 | 8.483e-01 | 4.130e-03 | 1.477e-19 |
| | | 2000 | 8.489e-01 | 3.068e-03 | 8.532e-01 | 3.350e-03 | 5.220e-08 |
| | | 3000 | 8.549e-01 | 2.966e-03 | 8.593e-01 | 2.800e-03 | 7.280e-08 |
| | | 4000 | 8.572e-01 | 3.749e-03 | 8.629e-01 | 1.749e-03 | 6.160e-06 |
| | T2GFormer | 1000 | 8.302e-01 | 6.270e-03 | 8.286e-01 | 1.756e-02 | 7.166e-01 |
| | | 2000 | 8.402e-01 | 6.429e-03 | 8.444e-01 | 7.657e-03 | 9.535e-03 |
| | | 3000 | 8.475e-01 | 5.003e-03 | 8.444e-01 | 1.111e-02 | 8.923e-01 |
| | | 4000 | 8.508e-01 | 5.679e-03 | 8.540e-01 | 3.901e-03 | 7.649e-02 |

Table 9: Predictive performance in the form of $R^2$, for six different datasets and five different models. We compare the performance when the graph is fully connected versus when it is pruned to its true edges only. The statistics $\mu$ and $\sigma$ represent the mean and standard deviation aggregated over seeds and cross validations. With the Linear Mixed Effects (LME) Model we test if the $R^2$ results obtained with the pruned graph are greater than those with the fully connected graph. See Figure 10 for the same results.

| | | Graph Statistic | Fully connected | | True edges only | | LME |
| Dataset | Model | $n_{\text{train}}$ | $\mu$ | $\sigma$ | $\mu$ | $\sigma$ | $p$-value |
|---|---|---|---|---|---|---|---|
| | TabPFN | 1000 | 8.721e-01 | 1.974e-03 | | | |
| | | 2000 | 8.756e-01 | 9.270e-04 | | | |
| | | 3000 | 8.780e-01 | 3.718e-04 | | | |
| | | 4000 | 8.783e-01 | 2.596e-04 | | | |
| | XGBoost | 1000 | 8.495e-01 | 4.047e-03 | | | |
| | | 2000 | 8.598e-01 | 1.057e-03 | | | |
| | | 3000 | 8.618e-01 | 7.716e-04 | | | |
| | | 4000 | 8.642e-01 | 7.016e-04 | | | |
| | BDgraph | 1000 | 7.968e-01 | 1.242e-03 | | | |
| | | 2000 | 7.973e-01 | 5.478e-04 | | | |
| | | 3000 | 7.972e-01 | 1.416e-04 | | | |
| | | 4000 | 7.974e-01 | 1.145e-05 | | | |
| SCM 2 | FTTransformer | 1000 | 7.988e-01 | 6.280e-03 | 8.068e-01 | 5.799e-03 | 2.428e-11 |
| | | 2000 | 8.069e-01 | 5.623e-03 | 8.118e-01 | 3.220e-03 | 2.246e-07 |
| | | 3000 | 8.122e-01 | 2.971e-03 | 8.158e-01 | 1.102e-03 | 2.046e-08 |
| | | 4000 | 8.137e-01 | 9.271e-04 | 8.171e-01 | 1.481e-03 | 4.620e-10 |
| | FiGNN | 1000 | 7.913e-01 | 1.540e-02 | 7.994e-01 | 1.198e-02 | 1.783e-04 |
| | | 2000 | 8.093e-01 | 5.427e-03 | 8.112e-01 | 6.110e-03 | 5.826e-02 |
| | | 3000 | 8.152e-01 | 3.812e-03 | 8.073e-01 | 1.318e-02 | 9.948e-01 |
| | | 4000 | 8.162e-01 | 1.393e-03 | 8.166e-01 | 1.864e-03 | 2.944e-01 |
| | INCE | 1000 | 7.922e-01 | 6.950e-03 | 8.021e-01 | 5.126e-03 | 7.559e-20 |
| | | 2000 | 8.033e-01 | 3.749e-03 | 8.054e-01 | 5.855e-03 | 4.815e-02 |
| | | 3000 | 8.068e-01 | 4.149e-03 | 8.137e-01 | 3.154e-03 | 2.383e-18 |
| | | 4000 | 8.069e-01 | 3.399e-03 | 8.132e-01 | 2.133e-03 | 2.872e-07 |
| | T2GFormer | 1000 | 7.863e-01 | 1.251e-02 | 7.923e-01 | 1.657e-02 | 2.163e-02 |
| | | 2000 | 7.974e-01 | 1.011e-02 | 8.084e-01 | 4.063e-03 | 1.483e-09 |
| | | 3000 | 7.993e-01 | 9.041e-03 | 8.115e-01 | 4.340e-03 | 2.233e-09 |
| | | 4000 | 8.020e-01 | 8.168e-03 | 8.125e-01 | 4.053e-03 | 1.372e-04 |
| | TabPFN | 1000 | 7.972e-01 | 1.963e-03 | | | |
| | | 2000 | 8.024e-01 | 1.394e-03 | | | |
| | | 3000 | 8.035e-01 | 1.564e-03 | | | |
| | | 4000 | 8.042e-01 | 1.431e-03 | | | |
| | XGBoost | 1000 | 7.725e-01 | 5.500e-03 | | | |
| | | 2000 | 7.858e-01 | 1.684e-03 | | | |
| | | 3000 | 7.908e-01 | 1.714e-03 | | | |
| | | 4000 | 7.896e-01 | 1.285e-03 | | | |
| | BDgraph | 1000 | 7.575e-01 | 1.307e-03 | | | |
| | | 2000 | 7.587e-01 | 3.103e-04 | | | |
| | | 3000 | 7.606e-01 | 4.838e-05 | | | |
| | | 4000 | 7.607e-01 | 1.319e-05 | | | |

## D.2 Higher number of training samples

We repeat the experiments from Section 4 while varying the number of training samples $n_{\text{train}}$ up to $10^5$ samples. Note that $10^5$ samples is quite high, considering the low number of features $p = 10$ and the relatively simple feature interactions in the datasets. To reduce computational cost, we do not tune the hyperparameters but use the default hyperparameters as previously described in Appendix C, and we only evaluate two of the six datasets. We do use the same cross-validation strategy with 10 seeds as before. The results for the learned graph structure are shown in Figure 11, results for the predictive performance are shown in Figure 12. The results are consistent with the main experiment (tuned hyperparameters, up to $n_{\text{train}} = 4000$ samples), shown in Figure 9 and Figure 10. However, the results have a larger variety because the hyperparameters are not tuned.

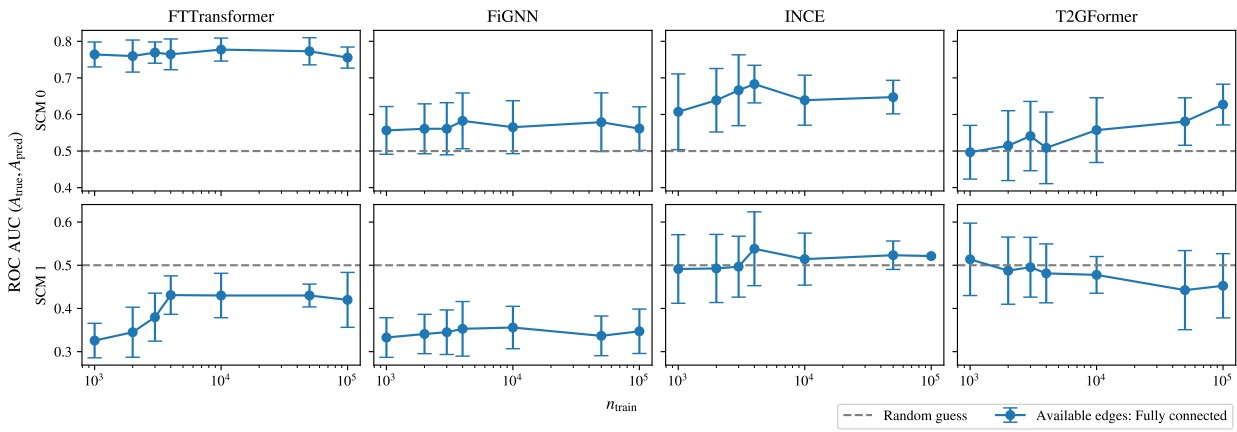

Figure 11: Graph quality in the form of the ROC AUC with a higher number of training samples $n_{\text{train}}$ than the main experiment, for two different datasets. Error bars show standard deviation across seeds and cross validations.

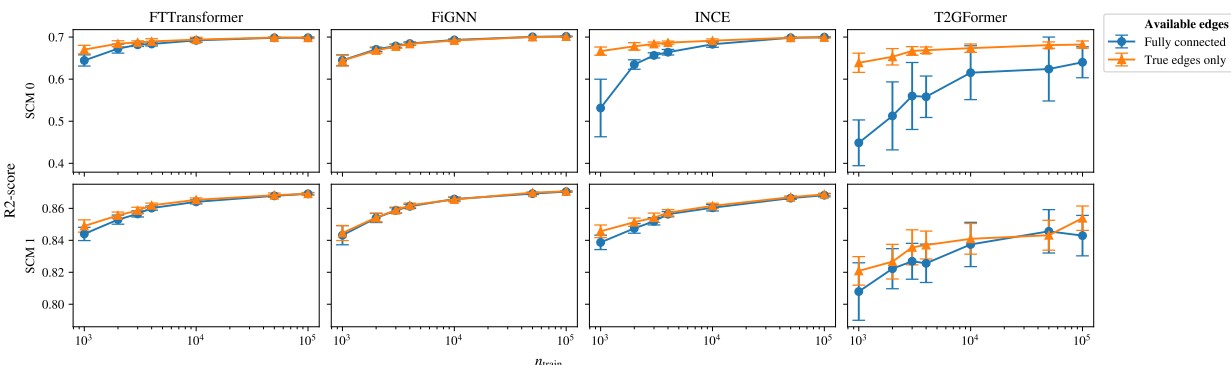

Figure 12: Predictive performance with a higher number of training samples $n_{\text{train}}$ than the main experiment, for two different datasets. Error bars show standard deviation across seeds and cross validations.

## D.3 Node-level versus graph-level models

In Figure 5, not all models benefit from pruning the graph. For FiGNN, the pruned and fully connected graphs have similar performance. This could be because FiGNN treats the task of predicting the target feature on a graph-level task, while the other models have a target token and treat it as a node-level task. As an example, we adapt the architecture from T2G-Former, which is by default a node-level model, to a graph-level model. Results on the predictive performance are shown in Figure 13. The graph-level model

benefits less from pruning the graph than the node-level model. This indicates that the node-level models are more sensitive to the graph structure than the graph-level models.

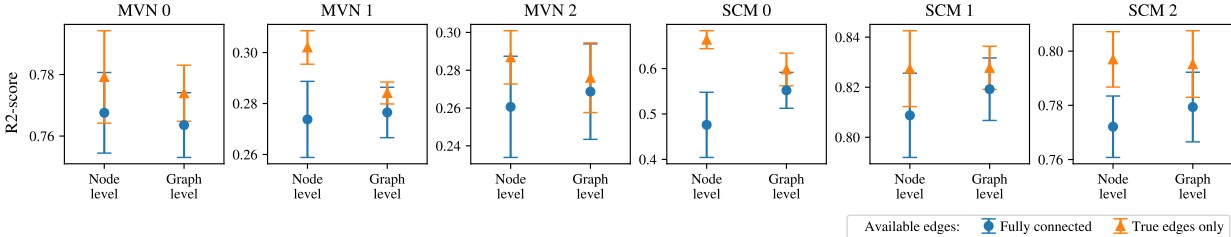

Figure 13: Node-level (default) versus graph-level adaptions of T2G-Former for multiple MVN and SCM datasets. The node-level adaptation benefits more from pruning the graph to the true edges. Error bars show standard deviation across seeds and cross validations.

