# OpenReview forum: "The Role of Feature Interactions in Graph-based Tabular Deep Learning"
_TMLR — Accepted by TMLR_

### Review · Reviewer_JABv · 2025-12-05

**Summary Of Contributions:**

The paper investigates the capability of Graph-based Tabular Deep Learning (GTDL) methods to learn feature interactions. The authors introduce a framework using synthetic datasets where the ground-truth feature graph is known. Through this, they demonstrate that existing GTDL methods (such as FiGNN and FT-Transformer) fail to recover meaningful feature interactions, yielding ROC AUC scores near random chance. Furthermore, the authors show that pruning the graph to include only true edges generally improves predictive performance, validating the hypothesis that structural fidelity drives accuracy.

**Additional Comments:**

I have enjoyed reviewing this work; I find it interesting and scientifically sound. I am willing to recommend this work for acceptance after the mentioned concerns are tackled.

**Audience:**

Yes

**Audience Explanation:**

The findings of this paper are likely to be of significant interest to the TMLR audience, particularly those working within the Graph Learning and Tabular Learning communities. The work addresses a notable gap in the existing literature by systematically benchmarking GTDL methods using synthetic data with known ground-truth feature graphs. This methodological contribution, combined with the paper's clear structure and the availability of code for reproducibility, makes it a valuable resource.

**Claims And Evidence:**

Yes

**Claims Explanation:**

The problem outlined is significant, and the motivation is scientifically plausible. The introduction flows well, and the proposed approach of using synthetic data to isolate structural learning is well-designed. The results are convincing, and the provision of code and a clear evaluation setup is commendable.

**Requested Changes:**

Major Comments
- **Claims regarding Data Symmetry** The statement in Section 4.2—"When data is scarce, symmetries in the data are useful. However, when data is abundant, the model does not have to rely on symmetries in the data"—is too bold and incorrect. This contradicts recent literature on data symmetry discovery (e.g., Marchetti et al., "Harmonics of Learning"). Symmetries remain important in both regimes; however, in large-scale data regimes, these symmetries are often learned implicitly rather than requiring explicit inductive bias. The authors should revise this to reflect that the abundance of data facilitates implicit learning, rather than rendering symmetries useless.

- **Exclusion of State-of-the-Art Baselines** The authors explicitly exclude tree-based models (XGBoost, LightGBM) and recent foundation models (TabPFN, TabICL, etc.). While I understand these methods do not explicitly represent graphs, excluding them contradicts the work's motivation. The paper aims to validate the impact of true feature interactions on downstream performance. To prove that explicit structure is beneficial, the proposed technique (GTDL with true graphs) must be benchmarked and discussed against SOTA tabular methods that model interactions implicitly.

- **Statistical Verification** Figure 5 outlines the main results regarding predictive performance. While the improvement from pruning the graph is visible, the paper lacks statistical rigor here. I strongly suggest adding statistical tests (e.g., t-tests or Wilcoxon signed-rank tests) to verify the significance of the performance gap between the "Fully connected" and "True edges only" settings in Section 4.2. Also, the inclusion of explicit tables would be beneficial for the community.

- **Clarity of the Motivating Example** The example provided in Section 3.1 ($x_0 - x_1 - x_2$) is confusing. The text argues that the model should instead recognize $x_1$ as a better predictor for $x_2$. It is unclear why $x_1$ is inherently better than $x_0$ for predicting $x_2$ without more context. I would ask the authors to clarify this example to make the argument clear and potentially provide other examples.


Minor Comments & Clarifications
- In Section 3.3/4.2, the authors state: “By restricting the model to only the true interactions, the learning process becomes more efficient and focused…”. What specifically is meant by a "more focused" learning procedure?
- While Figure 4 accurately outlines the claims, the visual presentation could be improved for better readability. (Suggestion only).

---

> ### Author Response · Authors · 2026-01-14
> **Response to Reviewer JABv (1/2)**
>
> We thank the reviewer for their supportive feedback. We acknowledge that the requested changes have improved the paper significantly. In the new version, we have responded to all comments of the reviewer and incorporated their suggestions. In the following, we discuss each point in detail.
>
> ## Claims regarding data symmetry
> We agree that our initial statement is too bold, and indeed meant that the model implicitly learns the symmetries from the data. We updated the sentence in Section 4.2 (p10, §4) to:
>
> > When data is scarce, explicitly incorporating the symmetries from the data into the model is beneficial. However, an abundance of data facilitates an implicit learning of these symmetries (Marchetti et al., 2024), reducing the benefit of explicitly incorporating them.
>
> ## Exclusion of SOTA baselines
> We have included baselines to compare the predictive performance. We added to Section 3.4 (p9, §2):
>
> > Finally, we include TabPFN and XGBoost as additional baselines given their overall strong performance, to understand the difficulty of the predictive task.
>
> The results of the predictive performance are included Section 4.2 (p10, §3), Figure 5 (p11), Figure 10 (p26) and Table 9 (p27).
>
> ## Statistical Verification
>
> We thank the reviewer for their helpful suggestion. We agree that adding both statistical test and including tables improves the results. Added to Section 4.2 (p10, §2):
>
> > We use the Linear Mixed-Effects Model (Lindstrom \& Bates, 1988; Pinheiro \& Bates, 2000) to assess whether the pruned graph outperforms the fully connected graph. We elaborate on the statistical testing procedure in Appendix C.1.
>
> Appendix C.1 (p23):
>
> > To assess the statistical significance of the predictive performance improvement when using the pruned graph compared to the fully connected graph, we used a Linear Mixed Effects Model (Lindstrom \& Bates, 1988; Pinheiro \& Bates, 2000). We selected this approach over aggregating results per fold (e.g., for a Wilcoxon signed-rank test (Wilcoxon, 1945)) to avoid the loss of statistical power, given the limited number of cross-validation folds $(4, 3, 2, 1)$ available for $n_\text{train} = (1000, 2000, 3000, 4000)$, respectively. Since we evaluate 10 random seeds per fold, the data exhibits a hierarchical structure. Treating these seeds as independent samples would violate the independence assumption of standard tests and lead to pseudoreplication (Nadeau \\& Bengio, 1999). Therefore, we modeled the cross-validation fold as a random effect to account for the correlation between seeds, and the graph type (pruned or fully connected) as a fixed effect. Although not documented in this work, we compared the Linear Mixed Effects Model with the one-sided Mann-Whitney U test (Mann \& Whitney, 1947) and found comparable $p$-values.
>
> The results shown in Figure 9 (p24) and 10 (p26) (graph quality and predictive performance per dataset) are now also visible in Table 8 (p25) and 9 (p27).
>
> ## Clarity of Motivating Example
> We thank the reviewer for asking clarification of the motivating example in Section 3.1 (p6, last §). We rewrote it to clarify why $x_1$ is inherently better than $x_0$ for predicting $x_2$:
>
> > Consider the simple example: $x_0 - x_1 - x_2$. The model could learn to use $x_0$ to predict $x_2$ directly by learning an edge between them. However, this is suboptimal because $x_2$ is conditionally independent of $x_0$ given $x_1$. That is, once $x_1$ is known, $x_0$ provides no additional information for predicting $x_2$ (Lauritzen, 1996). When observations are noisy, using $x_0$ to predict $x_2$ accumulates uncertainty across multiple dependencies, leading to worse predictions than those obtained by using the immediate neighbor $x_1$. The model should instead recognize $x_1$ as a more reliable predictor for $x_2$, which results in learning the correct graph structure.

---

> > ### Author Response · Authors · 2026-01-14
> > **Response to Reviewer JABv (2/2)**
> >
> > ## Meaning of focused
> > We agree that the use of 'focused' in the initial sentence is not clear. In the new version, we have updated this discussion in Section 4.2 (p9, last §) as follows:
> >
> > > Restricting the model to only the true interactions simplifies the optimization landscape, allowing the learning algorithm to focus on meaningful relationships rather than being distracted by false edges. This leads to better generalization and higher predictive accuracy. In contrast, fully-connected models must learn to ignore many false edges, which can introduce noise and make optimization more difficult, especially when data is limited.
> >
> > Quoting the review: "In Section 3.3/4.2, the authors state:". While the reference to Section 4.2 is clear, we kindly seek clarification regarding the reference to Section 3.3 in the review, as we were unable to fully understand its intended meaning. We would be grateful if the reviewer could elaborate on this point.
> >
> > ## Figure 4
> > Thank you for this suggestion. We have added a black-dashed line at 0.5 in each subplot to emphasize that all ROC AUCs are close to random chance. This has been added to Figure 9 as well.

---

### Review · Reviewer_gzJV · 2025-12-26

**Summary Of Contributions:**

This paper conducts an empirical study to determine whether established graph-based tabular models (GTDL methods) are able to discover the true feature interaction graph. To this end, the authors construct two graph generation mechanisms which model linear and non-linear interactions, respectively. Data generated from these mechanisms is then used to train GTDL methods and to determine how well these methods are able to recover the original graph structure. The authors also perform an ablation on substituting the randomly initialized complete graph with the true graph to determine the effect of discovering the true feature interactions on predictive performance.

### Weakness
This could be due to a misunderstanding on my part, but from what I understand, in GTDL, the interaction graph weights are learned from e.g., an attention matrix like computation before being fed into a GNN, or directly processed as part of a transformer. Why do the authors expect the models to learn the feature interaction graph in the first place? For example, the inverse of the interaction graph could give equally strong signal to a GNN (especially, if it can learn or make changes to the edge embeddings). In a transformer, I understand the intuition more clearly, since lower attention weights literally mean that less information is exchanged between features. However, GNNs could learn to pay more "attention" to low edge weights, and still have strong predictive power.

**Audience:**

Yes

**Audience Explanation:**

Given the strong interest in tabular deep learning (TDL), I am very certain that the findings in this paper are highly relevant, in particular to researchers aiming to improve graph-based TDL methods.

**Claims And Evidence:**

Yes

**Claims Explanation:**

To me the main claim is that the presented GTDL methods are unable to accurately determine the true underlying feature interactions. This claim has been convincingly supported by the main experiments. I agree with the authors that if the methods cannot determine the feature interactions in simple synthetic data, they are unlikely to be broadly applicable in the real-world. Results in Figure 4 are very convincing, essentially establishing that discovering the true structure is possible (at least in the linear case), but not by the GTDL methods.

Another claim is that providing the ground-truth feature interactions helps in terms of predictive performance, which is not as convincing, but still convincing enough. Specifically, the R2 scores with and without pruned edges are often within each other's standard deviation. And even if not, they are only marginally outside. I would have expected the gap to be much clearer, especially given the simple nature of these tasks. I'd appreciate any comments on this by the authors.

**Requested Changes:**

None of these requests are critical to my recommendation:

* Please formally introduce the GNN based methods for GTDL. You do not have to introduce them all, but it would be very useful to at least describe the overall strategy more formally. From Section 2.2 alone, it is not fully clear exactly why GNNs are necessary here, or how they can be distinguished from Transformer-based methods.
* The attention equation does not really serve a real purpose in the paper. The construction of Q, K, and V matrices in the context of GTDL isn't even specified. Instead of this equation, I would recommend to add more information to the main text on how the attention map is extracted.

---

> ### Author Response · Authors · 2026-01-14
> **Response to Reviewer gzJV**
>
> We thank the reviewer for their encouraging and constructive response. We agree with the reviewer's requested changes, although non-critical for their recommendation, and have incorporated both. Before we discuss those, we first address two others question raised in the review.
>
> ## Why should models learn a graph?
> We expect that GTDL models should learn the correct graph is because of two reasons. First, the authors of the evaluated GNN GTDL methods (e.g., FiGNN, INCE, T2GFormer) use the edge weights for interpretation of the learned feature interactions. To help us address the second reason more effectively, we would appreciate it if the reviewer could kindly clarify what is meant by "the inverse of the interaction graph." To assist in the meantime, we offer an explanation based on two possible interpretations.
>
> If we interpret the inverse of the interaction graph as the [transpose of a directed graph](https://en.wikipedia.org/wiki/Transpose_graph) (redirecting all edges from $(u, v)$ to $(v, u)$), then the answer is, indeed, models should learn to both use the original and the redirected edges. This is because parent nodes also contain information for child nodes to do prediction on. This is visualized in Figure 7 (p20). Not the DAG, but the moralized and marginalized DAG is used as the true graph. This makes the true graph structure symmetric.
>
> If we interpret the inverse of the interaction graph as switching all 0s and 1s in the adjacency matrix ($A_\text{new}= 1 - A_\text{original}$), then the answer is no, models should not learn an inverse graph. We agree with the intuition that a GNN, in general, could learn to use edge embeddings with small values rather than large values. However, due to the specific implementations of the GNN GTDL methods (T2GFormer, FiGNN, INCE), this does not apply to these methods. We explain it briefly for these three methods, all equations refer to the original papers, not to our submission.
>
> - **T2GFormer and FiGNN:** The edge embeddings of (T2GFormer and FiGNN) are being consolidated into a single graph representation (equations (4-6) in T2GFormer, equations (2-3) in FiGNN). This graph representation is being used as a weighted adjacency matrix (equation (8) in T2GFormer, equation (6) in FiGNN), and multiplied with node representations, similarly how attention multiplies $a V$, with the attention map $a$ and the value matrix $V$, to obtain the next representation. So, similar to transformers, it means that lower adjacency matrix values mean that less information is exchanged between features.
> - **INCE:** In INCE, the edge embeddings are being used to post-hoc calculate the strength of feature interactions (Section 6.2 in INCE). This post-hoc calculation does not use the magnitude (low/high) of the edge weights, but uses a statistical distance between the edge embeddings to compute the adjacency matrix. So if the GNN learns to pay more 'attention' to low edge weights, this does not necessarily mean that a low-weight edge would have a low value in the learned adjacency matrix.
>
> ## Why is the improvement in predictive performance not that large?
> As suggested by reviewer JABv, we have added statistical tests that show that for small number of training samples the improvement when pruning is statistically significant. However, we agree that gap between the pruned and the fully connected graph is smaller than one would expect. We discuss two explanations why the improvement could be smaller than expected.
>
> 1. In our work, we only enforce the correct graph structure. That is, only *if* features can directly interact. However, *how* features interact is not enforced. Given that the correct graph structure does not emerge in GTDL methods, it is unlikely that correct functional forms of the feature interactions (i.e. how features interaction), do emerge. If this correct functional form is also enforced or learned, the difference in R2 improvement might be higher. This is briefly discussed as the first direction of future work in Section 5 (p11, §3).
> 2. The tasks are, by design, quite easy. If the fully connected graph already performs good, it is harder for the pruned graph to improve a lot because of the performance saturation.

---

> > ### Author Response · Authors · 2026-01-14
> > **Response to Reviewer gzJV (2/2)**
> >
> > ## Formal intro of GNN
> > We have added the following to the first paragraph of Section 2.2 (p4, §1).
> >
> > > Generally, the message-passing mechanism can be summarized as $$h_i^{(l)} = \phi^{(l)} \left( h_i^{(l-1)}, \bigoplus_{j \in \mathcal{N}(i)} \psi^{(l)} \left( h_i^{(l-1)}, h_j^{(l-1)}, A_{ij} \right) \right),$$ with $h_i^{(l)}$ the representation of node $i$ at layer $l$, $\mathcal{N}(i)$ the neighbors of node $i$, \footnote{In GTDL, the neighbors are typically all other features, i.e., $\mathcal{N}(i) = \{1, \ldots, p\} \setminus \{i\}$. A fully connected graph is used due to the absence of a known graph structure.} $\psi^{(l)}$ a message function, $\bigoplus$ an aggregation operator, and $\phi^{(l)}$ an update function. The details of these differ between GNN architectures. Attention-based methods can be seen as a special case of GNNs (Joshi, 2025 "Transformers are Graph Neural Networks"). In the context of tabular data, a key advantage of GNNs is that they generalize attention-based methods by using a trainable weighted adjacency matrix, $A$, to explicitly propagate information between nodes—rather than relying on implicitly learned attention maps. Furthermore, GNNs for tabular data typically have trainable parameters that represent individual features or interactions, allowing for more flexible modeling of feature interactions compared to attention-based methods that rely on shared parameters across features. Therefore, we refer to them as *explicit* GTDL methods, contrary to attention-based methods that model the graph structure implicitly.
> >
> > ## Attention equation and interpretation
> > We agree that the full attention equation is unnecessary, considering the space it requires. We initially introduced this to emphasize what we mean with the attention map $a$. We removed the display/non-inline equation including its related sentences. Instead, we renamed Section 3.2 (p7) from "Metric for evaluating feature interactions" to "Interpreting and evaluating the graph structure". One of its added paragraphs (p7, last §) is about extracting the attention maps and interpreting them as the adjacency matrix:
> >
> > > After training, the attention maps of the test samples are extracted for all heads and layers. To obtain the adjacency matrix we perform two steps. First, we average the attention maps over the test samples, heads and layers to obtain and to obtain a single attention map of size $p \times p$. Second, we account for the diagonal of the attention map and for the softmax normalization from the original attention equation, which is further explained in Appendix A, to obtain the weighted adjacency matrix $A$.
> >
> > Section 5 contains an extra paragraph (p11, §2) discussing limitations of the above procedure. Furthermore, Section 3.2 has an extra paragraph (p7, §4) that gives an example to justify the interpretation of the attention map as the adjacency matrix.

---

### Review · Reviewer_bWh1 · 2025-12-29

**Summary Of Contributions:**

This paper investigates whether Graph-based Tabular Deep Learning (GTDL) methods—including both explicit feature-level GNNs and implicit attention-based architectures—actually learn meaningful feature-interaction graphs.
The authors' contributions include:
• A Systematic Evaluation Framework: A pipeline to generate synthetic datasets with known ground-truth interaction structures using multivariate normal (MVN) models and structural causal models (SCM).
• Quantitative Benchmarking: An evaluation using ROC-AUC to compare learned adjacency matrices against the ground-truth graph.
• Empirical Analysis: An assessment of several GTDL models, revealing that they currently fail to recover true interactions (performing near random guessing).
• Performance Insight: Evidence showing that manually imposing the true graph structure via pruning improves predictive performance, suggesting that current models suffer from an inability to learn the correct structural inductive bias.

**Audience:**

Yes

**Audience Explanation:**

Yes. The TMLR audience, particularly those working on Tabular Deep Learning and Graph Neural Networks, would find this work highly relevant.
The paper challenges a core assumption in the GTDL literature: that these models "learn" the underlying structure of tabular data. By providing a reproducible framework for verifying these claims, the authors encourage the community to move toward more rigorous evaluation standards and to develop models with more explicit structure-aware objectives.

**Broader Impact Concerns:**

There are no significant ethical or broader impact concerns for this work. The paper focuses on the fundamental evaluation of machine learning architectures using synthetic data.

**Claims And Evidence:**

No

**Claims Explanation:**

No, the evidence is currently only partially convincing. While the empirical findings clearly show that the tested models fail to recover ground-truth graphs, the conclusion that these models are inherently unable to learn such interactions is undermined by the experimental setup:
• Sample Size Limitations: The datasets used are relatively small (1,000 to 4,000 samples). Complex GTDL models may require significantly more data to infer intricate interaction patterns. This is corroborated by Figure 10 in the Appendix, which shows a performance trend upward as sample size increases. Without exploring larger data regimes, it is unclear if the failure is architectural or simply a result of data scarcity.
• Lack of Baselines: The absence of traditional ML baselines (e.g., XGBoost, Random Forest) makes it difficult to assess the difficulty of the task. If standard models also fail to capture interactions in these specific regimes, the issue might lie in the dataset complexity rather than the GTDL architectures themselves.
• Interpretability of Attention: The mapping of attention matrices to weighted adjacency graphs is a strong assumption. The paper lacks a rigorous justification for this mapping, which is critical given the ongoing academic debate regarding the interpretability of attention.

**Requested Changes:**

To meet the threshold for acceptance, the following changes are requested:
1. Large-Scale Experiments: Conduct experiments on significantly larger synthetic datasets to determine if the failure to learn interactions persists or if it is a function of sample size.
2. Baselines Comparison: Include traditional machine learning models (XGBoost, LightGBM, or Random Forests). Use their internal feature importance or interaction scores to provide a benchmark for the "learnability" of the ground-truth graphs.
3. Justification of Metrics: Elaborate on the rationale for treating attention weights as a proxy for adjacency, addressing potential limitations of this interpretation.

---

> ### Author Response · Authors · 2026-01-14
> **Response to Reviewer bWh1 (1/2)**
>
> We thank the reviewer for their thoughtful feedback. Below, we discuss the three requested changes.
>
> ## Large-Scale Experiments
> We thank the reviewer to raise the valid concern that failure to learn the correct graph structure and the difference between the predictive performance of the fully connected and pruned graph could be because of data scarcity. We have included extra experiments to validate this concern. Both in Section 4.1 (p9, §4) and Section 4.2 (p10, §4) we added a sentence:
>
> > This observation holds when increasing the number of training samples up to $10^5$ samples, discussed in Appendix D.2.
>
> Where 'this' in the previous sentence refers to the fact that for more number of training samples, the ROC AUC does not increase and that the difference of R2 scores between the fully connected and pruned graph become smaller. In the new Section D.2 "Higher number of training samples" (p31) we elaborate:
>
> > We repeat the experiments from Section 4 while varying the number of training samples $n_\text{train}$ up to $10^5$ samples. Note that $10^5$ samples is quite high, considering the low number of features $p=10$ and the relatively simple feature interactions in the datasets. To reduce computational cost, we do not tune the hyperparameters but use the default hyperparameters as previously described in Appendix C, and we only evaluate two of the six datasets. We do use the same cross-validation strategy with 10 seeds as before. The results for the learned graph structure are shown in Figure 11, results for the predictive performance are shown in Figure 12. The results are consistent with the main experiment (tuned hyperparameters, up to $n_\text{train} = 4000$ samples), shown in Figure 9 and Figure 10. However, the results have a larger variety because the hyperparameters are not tuned.
>
>
> ## Baseline comparison
> We appreciate the suggestion to include baselines, and we have added extra experiments to compare the predictive performance. We added to Section 3.4 (p9, §2):
>
> > Finally, we include TabPFN and XGBoost as additional baselines given their overall strong performance, to understand the difficulty of the predictive task.
>
> The results of the predictive performance are included Section 4.2 (p10, §3), Figure 5 (p11), Figure 10 (p26) and Table 9 (p27).
>
> Regarding the structural analysis, we carefully considered the reviewer's suggestion to include proxies of adjacency matrices for traditional ML baselines (e.g., XGBoost, RandomForest). We fully agree with the intent to benchmark whether the ground-truth graphs are actually "learnable", but faced two methodological challenges for traditional baselines:
>
> - The reviewer suggests using internal feature importances. However, these metrics primarily capture the strength of the relationship between a feature and the target ($x_i \rightarrow y$). A full adjacency matrix, by contrast, must capture the structural relationships between the features themselves (both $x_i \leftrightarrow y$ and $x_i \leftrightarrow x_j$), which standard feature importance does not provide.
> - The reviewer suggests using the model's interaction scores. Unlike GTDL methods, traditional baselines do not natively output interaction scores that map directly to a graph structure. While post-hoc techniques (like SHAP interaction values) exist, extracting a learnable graph from them is less straightforward and introduces additional layers of interpretation. We felt this would deviate from the paper's primary focus on the native structural claims of GTDL methods.
>
> We agree with the reviewer that a benchmark for the graph learnability is necessary, and have used the non-ML statistical method BDgraph as such a benchmark. As shown in Figure 4 (p10), BDgraph effectively learns nearly correct graphs for MVN datasets and outperforms GTDL methods on SCM datasets (despite having lower predictive performance). This fulfills the role of a robust structural baseline, proving that the ground-truth graphs are indeed learnable from the data, even if GTDL methods currently fail to capture them. We hope that these clarifications, combined with the strong structural performance of the BDgraph baseline, satisfactorily address the concern regarding the graphs learnability.

---

> > ### Author Response · Authors · 2026-01-14
> > **Response to Reviewer bWh1 (2/2)**
> >
> > ## Justification of Metrics (Attention interpretation)
> > We agree that rationale for treating attention weights as a proxy for adjacency should be elaborated on. We thank the reviewer for this suggestion, as we believe this improves the clarity of the paper. To Section 2.1 (p3, §6) we added:
> >
> > > While in natural language the size of attention map differs per input due to varying sequence lengths, in tabular data the number of features is fixed, making the attention map the same size as the adjacency matrix. This allows for an interpretation of the attention map as a proxy for the weighted adjacency matrix, which is further discussed in Section 3.2.
> >
> > We renamed Section 3.2 (p7) from "Metric for evaluating feature interactions" to "Interpreting and evaluating the graph structure". One of its added paragraphs (p7, §4) connects attention to the adjacency matrix:
> >
> > > To justify this interpretation, consider the following example: Let $a \in \mathbb{R}^{p \times p}$ be the attention map, where each element $a_{ij}$, denotes the attention weight from feature $i$ to feature $j$, as used by Vaswani et al. (2017). If feature $i$ is conditionally independent of feature $j$ given all other features, we have $A_{\text{true},ij} = 0$. The attention mechanism should learn during training to assign low attention weights $a_{ij}, a_{ji} \approx 0$, as feature $i$ does not rely on information from feature $j$ (and vice versa) for its representation. Contrary, if feature $i$ and feature $j$ are dependent, we have $A_{\text{true},ij} = 1$. The attention mechanism should learn to assign non-zero attention weights $a_{ij}, a_{ji} > 0$, as feature $i$ relies on information from feature $j$ (and vice versa) for its representation. In summary, with $A_{\text{true},ij} = 0$ we expect $a_{ij}, a_{ji} \approx 0$, and with $A_{\text{true},ij} = 1$, we expect $a_{ij}, a_{ji} > 0$.
> >
> > Section 3.2 has an extra paragraph (p7, last §) discussing practically how the attention map is translated to the adjacency matrix. Furthermore, Section 5 contains an extra paragraph (p11, §2) discussing limitations of the above:
> >
> > > A potential risk could be the interpretation of the attention map as a weighted adjacency matrix. First, the attention mechanism is not designed to specifically model feature interactions or for explainability. If attention can be used for explainability is an ongoing debate (Bibal et al., 2022; Lopardo et al., 2024), although this debate in the literature mainly focuses around natural language tasks. However, our interpretation for tabular data—that non-interacting features will exhibit low attention values while interacting features will manifest higher attention—is deliberately modest and pragmatic, rather than relying on attention as a fully explanatory tool. Second, while we acknowledge that aggregating attention across layers and heads might obscure certain signals, we find no strong evidence that alternative aggregation strategies would significantly improve graph recovery or alter the overall conclusions.
> >
> > Finally, we note that we interpret the attention map as a proxy for the adjacency matrix, but do not assume that they should be exactly the same, nor that attention directly be used for explainability. We use it to measure the ROC AUC. This metric is a relative measure, meaning that  the attention map (interpreted as a learned adjacency matrix) can still have a high ROC AUC while being poorly calibrated.

---

> > ### Comment · Reviewer_bWh1 · 2026-01-26
> > **Large scale experiment**
> >
> > I appreciate the authors’ effort to increase the number of training samples for some datasets; however, several aspects remain unclear to me. I understand that hyperparameter tuning can be time-consuming and that it may reasonably be omitted since it is not the main focus of the paper. Nevertheless, it is not clear which specific component makes large-scale experiments infeasible. Is the bottleneck related to data generation, model training, or something else?
> >
> > I also acknowledge that the datasets consist of only 10 features and that the number of samples is orders of magnitude larger. However, my concern relates to the number of parameters of the models. For instance, FT-Transformer typically has a number of parameters on the order of millions. Is the available number of samples sufficient to reliably train such a large model?
> >
> > From Figure 12, we can consistently observe that the performance gap between models with the correct enforced structure decreases as the number of training samples increases, which further strengthens my concern regarding data availability. Moreover, considering Figure 11, despite the comparable predictive performance between the “true edge” and the fully connected models, we observe that in most cases there is no corresponding improvement in the graph quality metric. In some cases, this metric is even lower than that of a random prediction. How can the models achieve similar predictive performance while seemingly failing to capture the underlying graph structure?

---

> > > ### Author Response · Authors · 2026-01-28
> > >
> > > We thank the reviewer for their continued engagement with our work and for raising these insightful follow-up questions. We answer the three questions below.
> > >
> > > ### Bottleneck of large-scale experiments
> > > The primary bottleneck making large-scale experiments infeasible is the combination of model training time and the rigorous hyperparameter tuning, which is the main reason why we trained up to $4000$ samples in our initial submission.
> > >
> > > With default hyperparameters a single model training run typically takes a few minutes on an NVIDIA A100. During tuning, with certain hyperparameters, a single model training run can take approximately an hour. Because we tune for 50 runs for each cross-validation, this could potentially lead to very long compute times. Therefore, the repeated model training during the hyperparameter tuning and cross-validations makes large-scale experiments expensive. Finally, the same interpretations and conclusions hold, regardless of whether we tune hyperparameters or increase the training set beyond $4000$ samples.
> > >
> > > ### Number of model parameters
> > > We understand the reviewer's concern that models like FT-Transformer might be over-parameterized for the datasets in our experiments. However, we believe the current setup is valid for three reasons:
> > >
> > > - **Adaptive model complexity:** Let's consider the number of model parameters of FT-Transformer with 10 features, using the hyperparameter space and default as described in Table 3 (p22). This setup has minimum around 600, maximum 4.5 million model, and with the default setting 400.000 parameters. When the hyperparameters are tuned for 50 runs, the number of model parameters can adapt to the data size, favoring smaller architectures if necessary.
> > > - **Empirical stability:** The standard deviation when averaging over seeds and cross-validations (shown as the error bars in e.g. Figure 10 and 12) is relatively small compared to the averages themselves. When the number of samples would not be sufficient, one would expect a much larger variance in the performance across seeds and cross-validations. Furthermore, performance converges as $N$ approaches to $10^5$. This stability indicates that the combination of sample sizes and number of model parameters used in our experiments are adequate for the models to converge to their optimal performances.
> > > - **Real-world relevance** While datasets evaluated in our experiments can be considered small, their sizes are representative of the tabular domain. In the recent TabArena [1] benchmark, 20% of the datasets have $\leq 10$ features and $\leq 4000$ samples (rising to 30% for $\leq 10^5$ samples), see Figure 3 of [1]. Deep learning models are frequently applied to these "small" regimes in practice, making it crucial to understand their behavior here.
> > >
> > > ### How similar predictive performance when failing graph recovery?
> > > We agree with both reviewer's observations. When the number of training samples increases i) the performance gap between models with the correct enforced structure decreases, and ii) there is no corresponding improvement in the graph quality metric. Both observations have been discussed in  Section 4.2 (p10, §5) and Section 4.1 (p9, §4), respectively.
> > >
> > > The reviewer raises an excellent question: *"How can the models achieve similar predictive performance while seemingly failing to capture the underlying graph structure?".* While we do not have a definitive answer to this, we have implicitly touched upon this question in Section 5 (p10, §1), the conclusion:
> > > > This indicates that the mechanisms of message-passing in GNNs, and attention in transformers, does not work as intended for tabular data.
> > >
> > > And along similar lines in the abstract:
> > > > This suggests that the attention mechanism and message-passing schemes used in GTDL do not effectively capture feature interactions.
> > >
> > > Because the models are only incentivized to optimize predictive performance, they are not explicitly encouraged to capture the underlying causal or relational structure. However, this does not prevent the models from achieving good performance, due to the flexibility and expressiveness of deep learning architectures. The SGD optimization finds alternative pathways, potentially exploiting spurious correlations, redundant feature combinations, or distributed representations, that circumvent the need for accurate graph structure recovery while still achieving competitive predictions.
> > >
> > > Our claim is that, in the low-data regime, enforcing the true graph structure acts as a beneficial inductive bias: it constrains the model to learn through the correct feature interactions, leading to improved generalization. As the number of samples increases, the models have sufficient data to compensate for their structural misspecification, which explains why the performance gap diminishes while graph recovery remains poor.
> > >
> > > ---
> > >
> > > \[1\] Erickson et al. *TabArena: A Living Benchmark for Machine Learning on Tabular Data* (2025) https://arxiv.org/pdf/2506.16791

---

### Decision · Action_Editor_L1y4 · 2026-02-12

**Recommendation:** Accept as is

**Audience:**

Yes

**Audience Explanation:**

TMLR has an ample community working on graph representation learning, and this paper directly addresses one very common claim related to these models.

**Claims And Evidence:**

Yes

**Claims Explanation:**

The paper introduces a benchmark to generate tabular datasets having specific graph-based interactions between features. They show that existing graph / transformer models fail to learn this interaction graph, contradicting a common claim found in the literature.

All reviewers were generally positive of the paper. In particular, after the rebuttal they all agree the claims of the paper are interesting, the methodology is sound, and the conclusions are reasonable. Most of the concerns during rebuttal (lack of specific baselines such as TabPFN, limited scale) have been addressed during the rebuttal.

One reviewer remains concerned of the claim that "pruning the graph" to match the known interactions is always beneficial, and that the current scale of the experiments is enough to claim the results are always valid. However, they view the paper as a "valuable starting point for a necessary discussion within the community" and also technically sound.